

# Quantum error correction and large $N$

**Alexey Milekhin⋆**

Physics Department, Princeton University, Princeton, NJ

⋆ milekhin@ucsb.edu, maybe.alexey@gmail.com

## Abstract

In recent years quantum error correction (QEC) has become an important part of AdS/CFT. Unfortunately, there are no field-theoretic arguments about why QEC holds in known holographic systems. The purpose of this paper is to fill this gap by studying the error correcting properties of the fermionic sector of various large $N$ theories. Specifically we examine $SU(N)$ matrix quantum mechanics and 3-rank tensor $O(N)^3$ theories. Both of these theories contain large gauge groups. We argue that gauge singlet states indeed form a quantum error correcting code. Our considerations are based purely on large $N$ analysis and do not appeal to a particular form of Hamiltonian or holography.

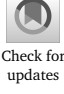

# 1 Introduction

## 1.1 Motivation

Seminal paper [1] by Almheiri, Dong and Harlow demonstrated that quantum error correction(QEC) naturally emerges in the AdS/CFT correspondence. The idea is simple: the same bulk region can be reconstructed using different parts of the boundary. So if some part of the boundary is lost or subject to quantum noise, information in the bulk is perfectly preserved and can be recovered using a different part of the boundary. This has lead to a variety of interesting results, such as entanglement wedge reconstruction [2] and derivation of Ryu–Takayanagi formula [3].

    Several toy models [4], [5] of error-correcting bulk-boundary correspondence were constructed using perfect and random tensor networks in the bulk. In these examples the boundary has one spatial dimensional and the bulk is two-dimensional Poincare disk. One drawback of these models is that they do not have a Hamiltonian, so they are not dynamical. These constructions resemble approximate wave function constructions for quantum many-body systems

using matrix product states(MPS) [6,7], projected entangled pairs(PEPS) [8,9], tree tensor networks [10] and multi-scale entanglement renormalization ansatz(MERA) [11]. However, it is not clear how they are related to conventional holographic systems, such as $\mathcal{N} = 4$ super Yang–Mills or at least Sachdev–Ye–Kitaev(SYK) model [12–15].

*The purpose of this paper is two-fold.* The first goal is to examine quantum error correcting properties in more standard holographic systems. As was anticipated and as we will see shortly, this quantum error correction is only approximate in the large $N$ limit. The second goal is to see how small this error can be. Recently it was proven that in the presence of continuous global symmetries there is a *lower bound* on recovery error [16], so the recovery cannot be perfect. In this paper we will prove an *upper bound* on error, thus demonstrating that it has to be small in the large $N$ limit. Let us describe these two goals in more detail.

Before considering a full quantum field theory problem, it is natural to study first quantum mechanical systems. In this paper we study generic $SU(N)$ matrix models and $O(N)^3$ tensor models. As an example, one can keep in mind Banks–Fischler–Shenker–Susskind(BFSS) [17] matrix model or Carrozza–Tanasa–Klebanov–Tarnopolsky (CTKT) [18, 19] tensor quantum mechanics. BFSS is a dimensional reduction of $\mathcal{N} = 4$ super Yang–Mills to quantum mechanics and it provides some of the strongest evidences for the holographic correspondence, as the gravity predictions from black hole thermodynamics have been matched with numerical Monte–Carlo simulation of this model [20–22]. CTKT model has the same large $N$ limit as SYK, but does not have any disorder.

However, we would like to emphasize once again that our results are based on certain large $N$ properties, and are not tied to any particular Hamiltonian. This may sound surprising for the following reason. In the original paper [1] on QEC in AdS/CFT the error correcting properties were tightly bound with bulk locality. It is true that BFSS at low energies does have a description in terms of ten-dimensional supergravity, but we do not expect bulk locality for a generic matrix quantum mechanics. However, it has been argued [23–26] that various large $N$ matrix models, even a harmonic oscillator, do have a description in terms of string theory. The corresponding geometry has a string size curvature. Our results suggest that error correction may be a generic feature of string theories.

Investigating QEC in generic matrix/tensor has one important drawback: they do not have a spatial structure. Nonetheless, in the spirit of holographic quantum error correction we can ask: *how robust are the code states in these models against erasures of fermions?* The Hilbert space of scalar fields is infinitely-dimensional, so we will postpone the investigation of the scalar sector to future work.

What is the error correcting code subspace in matrix/tensor models? It was suggested earlier [27] that the redundancy in holographic QEC maybe tied[1] the gauge redundancy. As another motivation, let us recall that both BFSS and CTKT(or SYK) have similar features: in both of them classical gravity description is expected to arise at low energies and large $N$ and they both have large internal symmetry groups. In the case of BFSS it was conjectured [28] that $SU(N)$ non-singlet in BFSS are gapped and therefore absent from low-energy gravity description. This statement was later corroborated by Monte–Carlo simulations [29]. Similarly, it was conjectured that $O(N)^3$ singlets form holographic states in CTKT model [30]. *Therefore, we are going to study error correcting properties of singlet states in large N matrix/tensor models.* Despite focusing on $SU(N)$ or $O(N)^3$ singlets, the states might be charged under other groups.

This brings us to the second goal of this paper. This interpretation of AdS/CFT also fits well with the expectation that the full theory of quantum gravity does not have any internal continuous global symmetries [31–33]. It has been known for a long time that quantum error correcting codes do not allow the presence of continuous global symmetries [34]. How-

---

[1]More specifically, the toy model of ref. [27] builds precursor operators on the gauge-invariant part of the Hilbert space.

ever, this statement, known as Eastin–Knill theorem, has a few crucial assumptions. The most important assumption for the applications in QFT and quantum gravity is that the quantum systems are *finite dimensional*. Obviously this is not the case in QFT. Also, the theorem states that *exact* quantum error correction is not possible. Recent papers have proven numerous bounds on *approximate* error correction [35, 36] in presence of continuous symmetries [16] and Haar-random charged systems [37]. Very roughly, one of the bounds states [16] that the recovery error[2] can not be smaller than

$$\text{error} \gtrsim \frac{Q}{n}, \tag{1.1}$$

where $Q$ is total charge of the original state and it is assumed that the system is built from $n$ elementary physical subsystems. For singlet states $Q = 0$, so this bound does not rule out perfect error correction. Moreover, we should mention right away that this bound is not directly applicable for our case, since the action of the symmetry group is not transversal.[3]

Notice that eq. (1.1) has the total number of degrees of freedom $n$ in the denominator. So in large systems the error can be small. *How small could it be?*

Despite being quantum mechanical systems, matrix/tensor models have a huge number of degrees of freedom in the large $N$ limit.[4] Our second goal is to show that there is an *upper bound* for recovery error in these models as long as the number of erasures is not too big.

Last, but not least, let us point out a resemblance between gauge singlets and the so-called *stabilizer codes* in quantum information theory. Code states in these codes are build as invariant states under a certain subgroup $\{G_i\}$ of the full Pauli group:[5]

$$G_i|c\rangle = |c\rangle. \tag{1.2}$$

In the present paper we essentially studied a version where stabilizer group is a *continuous* gauge group, with gauge group charges $Q_i$ annihilating the code states:

$$Q_i|c\rangle = 0. \tag{1.3}$$

There have been proposals to make decoherence–free subsystems [38] and scar–states[6] [39] using group charges $Q_i$. However, in the current paper we will study the QEC properties and the action of generic operators.

## 1.2 An illustration

Let us illustrate the problem we are addressing. Suppose we have a set of Majorana fermions $\psi_{ij}^a$ in the adjoint representation(indices $ij$) of $SU(N)$. For our purposes index $a$ can be treated as "flavor" index. In principle we can have some other fields as well. Imagine that all the states we are interested in are $SU(N)$ singlets. We start from a singlet state $|s\rangle$ and add non-singlet

---

[2]Here we simplify terminology for readers not familiar with quantum error correction. Technically, "error" in the above equation is how much fidelity $F$ is different from 1. If the state before applying errors is a pure state $|s\rangle$, and after applying errors and performing recovery it is mixed state $\rho$, then $F^2 = \langle s|\rho|s\rangle$.

[3]Transversal means that the system can be separated into several parts such that: the symmetry group do not mix the parts and any error involving a single part can be corrected. In our case we do not have a spatial structure, so all fields live in one "dot".

[4]Note that $n$ in the above bound is roughly the number of qubits, and not the dimension of the full Hilbert space. So $n \sim N^2$ for matrix models and $n \sim N^3$ for tensor models. So we do not expect that the error is exponentially small.

[5]For a single qubit, Pauli group is a discrete group generated by four Pauli matrices $\mathbf{1}, X, Y, Z$. So that in total there are 16 elements, as we should include matrices multiplied by $-1$ or $\pm i$. For bigger number of qubits one should take a direct product of Pauli groups for individual qubits.

[6]States which do not thermalize.

perturbations. They can be seen either as "errors" or excitations deliberately added by an observer. Specifically consider two states:

$$|\xi_1\rangle = \psi^1_{21}|s\rangle \quad \text{and} \quad |\xi_2\rangle = \psi^1_{23}\psi^1_{31}|s\rangle. \tag{1.4}$$

*Is there a measurement which can distinguish $|\xi_1\rangle$ and $|\xi_2\rangle$?* If there is, then we can act by another non-singlet operator $\psi$ and return back to the original state $|s\rangle$, since Majorana fermions square to one. In other words, we could say that these two types of errors are correctable.

In general, the answer is *no*. The fundamental fact of quantum mechanics is that states can be distinguished(with probability 1) by a measurement only if they are orthogonal. In the above example, the overlap $\langle\xi_1|\xi_2\rangle$ is not zero even if we take into account the singlet condition as we can form a singlet from three $\psi^1$:

$$\langle\xi_1|\xi_2\rangle = \langle s|\psi^1_{12}\psi^1_{23}\psi^1_{31}|s\rangle = \frac{1}{N^3}\langle s|\operatorname{Tr}\left(\psi^1\psi^1\psi^1\right)|s\rangle + [1/N \text{ terms}]. \tag{1.5}$$

Since $|s\rangle$ is a singlet, we kept only the singlet channel[7] in the product of three $\psi^1$:

$$\psi^1_{ij}\psi^1_{kl}\psi^1_{pq} = \frac{1}{N^3}\left(\delta_{jk}\delta_{lp}\delta_{iq} - \delta_{jp}\delta_{qk}\delta_{li}\right)\operatorname{Tr}\left(\psi^1\psi^1\psi^1\right) + [\text{non-singlets and } 1/N \text{ terms}]. \tag{1.6}$$

Factor[8] $N^3$ comes from putting $j = k$, $l = p$, $i = q$ in both sides and summing over $kpq$. Also we have not specified anything about the flavor index, so in general we do not expect that this matrix element is zero.

However, notice that there is a factor of $1/N^3$. If $N$ is large and we manage to show that the matrix element is small compared to $N^3$, it would imply that we can distinguish the above two states with probability which is close to 1. This way we would have *approximate* error correction.[9]

Naively, the expression $\operatorname{Tr}\left(\psi^1\psi^1\psi^1\right)$ contains $N^3$ terms and if Majorana fermions square to one, we expect that the corresponding matrix element will be of order $N^3$. In this paper we are going to show that one can bound this matrix by

$$|\langle s|\operatorname{Tr}\left(\psi^1\psi^1\psi^1\right)|s\rangle| \le 2N^{5/2}, \tag{1.7}$$

hence

$$|\langle\xi_1|\xi_2\rangle| \le \frac{2}{\sqrt{N}}. \tag{1.8}$$

So we are saved by a factor of $\sqrt{N}$ and in the large $N$ limit states $|\xi_{1,2}\rangle$ are indeed almost orthogonal, so the two errors can be corrected with very high probability.

Interestingly, scaling $N^{5/2}$ is what one expects from 't Hooft scaling which we review in Appendix D. This scaling is expected to hold in low-energy sector of BFSS matrix model. However, in the present paper we are going to derive similar bounds for *any* singlet states, not just for low-energy ones. Also for more complicated operators our bounds will be less strict than 't Hooft scaling.

It is known [40] that generalized coherent states in vector/matrix models are orthogonal in the large $N$ limit. It implies that their dynamics is classical in the large $N$ limit. However, these coherent states are singlets under the gauge group, whereas we are discussing non-singlets.

---

[7]Terms with $\operatorname{Tr}\left(\psi^1\right)\operatorname{Tr}\left(\psi^1\psi^1\right)$ are absent since $\psi^1$ is traceless, being $SU(N)$ adjoint.

[8]In the above equations there are $1/N$ corrections, which we will discuss later. They arise because delta-function in eq. (1.6) are not orthogonal and do not respect traceless condition.

[9]We are comparing the overlap $\langle\xi_1|\xi_2\rangle$ to one because with the proper normalization of Majorana operators $\langle\xi_1|\xi_1\rangle = 1$.

## 1.3 Outline of the paper

Most of this paper is devoted to showing that this trend continues to hold: there is indeed approximate error correction in the large $N$ limit as long as there are not too many errors. Essentially this will follow from the fact that *generic non-singlet states are orthogonal in the large N limit.* We are going to introduce the formalism of error operators and explain why correcting certain errors is equivalent to correcting erasures of fermions.

First we will have to generalize the bound (1.7) for more general operators and tensor models. Our main tool is the observation that we can bound operators using elementary $SU(N)$ representation theory. By repeatedly using Cauchy–Schwartz inequality and cutting and gluing operators we can bound arbitrary fermionic operators by certain quadratic Casimirs of auxiliary $SU(N)$.

For example, we can introduce $SU(N)_1$ which rotates only $\psi_{ij}^1$. Then operator $\mathrm{Tr}\left(\psi^1\psi^1\psi^1\psi^1\right)$ is proportional to quadratic Casimir of $SU(N)_1$:

$$\mathrm{Tr}\left(\psi^1\psi^1\psi^1\psi^1\right) \propto C_2\left(SU(N)_1\right). \tag{1.9}$$

Singlets under $SU(N)$ are not necessarily singlets under $SU(N)_1$. However, because of Pauli principle we can not have arbitrary "large" $SU(N)_1$ representations, so that the Casimir can be bounded[10] by $\propto N^3$. More precisely, using $\psi$ anti-commutation relations and cyclicity of the trace, one can easily obtain that it is proportional to identity operator:[11]

$$\mathrm{Tr}\left(\psi^1\psi^1\psi^1\psi^1\right) = \left(N^2-1\right)\left(2N-\frac{1}{N}\right)\mathbf{1} \le 2N^3 \times \mathbf{1}. \tag{1.10}$$

For both matrix and tensor models we find that the error can be bounded by a power of $1/N$:

$$\mathrm{error}_{\mathrm{matrix}} = 1 - F_{\mathrm{matrix}}^2 \lesssim \mathcal{O}\left(\frac{S_0}{N^{2/5}}\right),$$

$$\mathrm{error}_{\mathrm{tensor}} = 1 - F_{\mathrm{tensor}}^2 \lesssim \mathcal{O}\left(\frac{S_0^5}{N^2}\right), \tag{1.11}$$

where $S_0$ is the number of affected fermions. However, this set will be a bit different in matrix and tensor models: in the matrix case we allow up to $S_0 \lesssim N^{1/10}$ erasures, whereas in the tensor case we can allow up to $S_0 \lesssim N^{1/6}$ errors. These powers of $N$ do not look very impressive, but in many places we used very generous upper bounds, so the above powers of $N$ can probably be easily improved. Also, from the illustration above it is clear that error-correcting properties are going to hold as long as the number of affected fermions is parametrically smaller than $N$. However, in matrix models the number of singlet operators build from $k$ fermions grows exponentially [41] with $k$, whereas in tensor models this growth is even factorial [42], so naively one might expect that gauge-singlet codes can correct only $\sim \log N$ errors. So it will require some effort to show correctability of $\sim N^{\#}$ errors.

The idea of our derivation is elementary. We will borrow a simple formalism of trace-preserving quantum operations from quantum information theory. This formalism can describe erasures and more general quantum noise. Knill–Laflamme(KL) [43,44] condition states that the set of error operators $\{E_\alpha\}$ is correctable if and only if:

$$P_{\mathrm{code}}E_\alpha^\dagger E_\beta P_{\mathrm{code}} \propto P_{\mathrm{code}}\delta_{\alpha\beta}, \tag{1.12}$$

where $P_{\mathrm{code}}$ is the projector onto the code subspace. In our case $P_{\mathrm{code}}$ is the projector on singlets. This means that we are interested in the singlet channel in the product $E_\alpha^\dagger E_\beta$. Had it

---

[10]Again, this coincides with 't Hooft scaling.

[11]The author is grateful to I. Klebanov for this remark.

contained only the identity operator, it would have led directly to (1.12). However, in general we will also have a non-trivial singlet operator $\mathcal{O}_{\alpha\beta}$:

$$P_{\text{code}}E_\alpha^\dagger E_\beta P_{\text{code}} \propto P_{\text{code}}\delta_{\alpha\beta} + \mathcal{O}_{\alpha\beta}P_{\text{code}}. \tag{1.13}$$

*We show that the singlet channel $\mathcal{O}_{\alpha\beta}$ is suppressed by $1/N$.*

Secondly, it is a long way between $1/N$ term in the KL condition (1.13) and the error bound (1.11). The meaning of the original KL condition is that errors are orthogonal on the code subspace. It means we can come up with a measurement telling us with probability 1 which error has occurred(syndrome measurement) so that we can fix it. In our case errors will be almost orthogonal:

$$V_{c_2,\beta}^{c_1,\alpha} = \langle c_1|E_\alpha^\dagger E_\beta|c_2\rangle \approx \delta_{c_1 c_2}\delta_{\alpha\beta}, \ |c_{1,2}\rangle \in \mathcal{C}. \tag{1.14}$$

Therefore for a syndrome measurement we will have to find the nearest orthogonal basis. This problem is known in mathematics as orthogonal Procrustes problem, which we will discuss in Appendix A.3. It has a very simple explicit solution in terms of the square root of the above Gram matrix:

$$\left(\sqrt{V}\right)_{c_2,\beta}^{c_1,\alpha}. \tag{1.15}$$

So the existence of an effective syndrome measurement is tied to bounding various norms of $V$. The construction we present is known as "pretty good measurement" [45–48], which we tailor to large $N$ situation.

The paper is organized as follows. In Section 2 we briefly review the motivation why quantum error correction emerges in AdS/CFT. Section 3.1 is dedicated to our main technical tool: trace-preserving quantum operations. We explain in details what kind of quantum operations we will be using and state Knill–Laflamme theorem which gives necessary and sufficient conditions for a perfect error correction. In Section 3.2 we explain that erasure of a subsystem, commonly discussed within the AdS/CFT, can be described as a special quantum operation.

In Section 4 we discuss the QEC properties of $SU(N)$ matrix quantum mechanics. Section 5 is devoted to rank-3 tensor models. We describe the models and state our main results. Also we explain why error correction eventually breaks down.

In Conclusion we summarize our results and discuss numerous open questions.

Appendices are dedicated to technical proofs. In Appendix A we prove a general bound on recovery fidelity using some mild assumptions on singlet operator spectrum. In Appendix B we prove the necessary bounds for matrix models and in Appendix C we prove similar bounds for tensor models. In Appendix D we explain what we mean by "'t Hooft scaling" and give some evidence why it should hold in the low-energy sector of BFSS.

## 2 Holographic error correction

In this Section we very briefly review the motivation for holographic error correction following the original paper [1]. Results from this Section will not be used in the rest of the paper.

The motivation comes from HKLL AdS-Rindler bulk reconstruction [49,50]. HKLL provides an explicit formula for bulk fields in the leading order in $1/N$. Namely, studying free field equations of motion in Rindler wedge of $AdS_d$ one finds that one can reconstruct the bulk fields using only the data on the boundary of $AdS$. More precisely, given a boundary region $A$, one can reconstruct the bulk fields in the *casual wedge* $\mathcal{W}_C[A]$ [51,52] of $A$.

Consider a Cauchy slice of $AdS_3$ and separate the boundary into three regions, $A, B$ and $C$ - Figure 1.

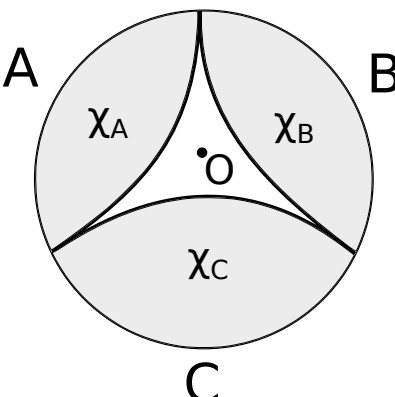

Figure 1: Bulk Cauchy slice of $AdS_3$ and three boundary regions $A, B, C$. Gray regions are the intersections of the corresponding causal wedges $\mathcal{W}_C$ and the Cauchy slice. Bulk fields within them can be reconstructed using the respective boundaries via HKLL. Bold lines $\chi$ are the causal surfaces. For empty $AdS$, $\chi$ actually coincides with Ryu–Takayanagi surface. However, in general $\chi_A$ is larger [52].

We see that point $O$ can not be reconstructed given any *single* region $A, B$ or $C$. However, it can be reconstructed from any *pair $AB$, $AC$* or *$BC$*. Therefore, if any of the three regions is lost(erased) we can still reconstruct point $O$. This is exactly the setup of quantum error correction.

As we will see in the next Section, it would be more convenient to study the protection against more general errors and derive the protection against erasure as a by-product.

# 3 Error operators formalism

## 3.1 Quantum operations

In this Subsection we will introduce the formalism of error operators which is standard in quantum information theory. We will describe the type of errors we will consider later in the paper. In the next Subsection we will argue that this set is enough to describe erasures at known locations.

How do we describe noise in a quantum system? [44] We can consider the situation when the original system, with density matrix $\rho$ is coupled to an environment in a state $|e_1\rangle$. Without loss of generality we consider the environment in a pure state. Then together they undergo a "normal" unitary evolution:

$$\rho \otimes |e_1\rangle\langle e_1| \rightarrow U \left(\rho \otimes |e_1\rangle\langle e_1|\right) U^\dagger. \tag{3.1}$$

Suppose that the environment has orthonormal basis $|e_\alpha\rangle, \alpha = 1, \ldots, M_0$. Then tracing out the environment yields:

$$\operatorname*{Tr}_{\text{Env}} \left(U\rho \otimes |e_1\rangle\langle e_1|U^\dagger\right) = \sum_{\alpha=1}^{K} \langle e_k|U|e_1\rangle\rho\langle e_1|U^\dagger|e_\alpha\rangle. \tag{3.2}$$

Introducing operators $E_\alpha = \langle e_1|U|e_\alpha\rangle$ we see that a generic quantum operation can be described by a set of operators $E_\alpha, \alpha = 1, \ldots, M_0$ acting on the system's density matrix $\rho$ as:

$$\rho \rightarrow \widetilde{\rho} = \sum_\alpha E_\alpha \rho E_\alpha^\dagger. \tag{3.3}$$

Imagine that the above operation is an undesirable noise and we want to correct it: come up with another operation(a set of $R_\mu$) restoring the original $\rho$:

$$\widetilde{\rho} \to \rho = \sum_\mu R_\mu \widetilde{\rho} R_\mu^\dagger. \tag{3.4}$$

Suppose we have a code subspace and a projector operator $P_{\text{code}}$ onto this subspace. One simple theorem from the quantum information theory is the following:

**Theorem 1.** *(Knill–Laflamme) [43, 44]: Quantum operation (3.3) can be corrected if and only if there is a complex matrix $N_{\alpha\beta}$ such that*

$$P_{\text{code}} E_\alpha^\dagger E_\beta P_{\text{code}} = N_{\alpha\beta} P_{\text{code}}. \tag{3.5}$$

So far we have not specified what kind of operators $E_\alpha$ we are considering, $\alpha$ is just an index to enumerate them. We will study $E_\alpha$ consisting of a single product(no contractions) of Majorana fermions $\psi_I$, where $I$ is could be a multi-index:[12]

$$E_\alpha = \prod_{I_l \in \alpha} \psi_{I_l}. \tag{3.6}$$

For example,

$$E_1 = \psi_1 \psi_2 \psi_{37},$$
$$E_2 = \psi_3 \psi_4 \psi_5 \psi_7 \psi_9. \tag{3.7}$$

There are two reasons for that. Majorana fermions $\psi_I$ can be represented as spins using Jordan–Wigner representation:

$$\psi_{2I-1} = Z \otimes \cdots \otimes Z \otimes X \otimes \mathbf{1} \otimes \cdots \otimes \mathbf{1}, \, X \text{ at position } I, \tag{3.8}$$

$$\psi_{2I} = Z \otimes \cdots \otimes Z \otimes Y \otimes \mathbf{1} \otimes \cdots \otimes \mathbf{1}, \, Y \text{ at position } I, \tag{3.9}$$

where $X, Y, Z$ are Pauli matrices. With this normalization, all Majorana fermions square to one:

$$\psi_I \psi_I = \mathbf{1}. \tag{3.10}$$

Therefore products of $\psi_I$ form conventional quantum noise operators such as bit flip $X$ and phase flip $Z$. More importantly, we will show in the next section that this is enough to describe erasures of subsystems.

Another reason is that we want to preserve the trace in the evolution (3.3). We can ensure that by requiring

$$\sum_\alpha E_\alpha^\dagger E_\alpha = \mathbf{1}. \tag{3.11}$$

If $E_\alpha$ is a simple product of $\psi_I$(up to a constant), then $E_\alpha^\dagger E_\alpha \propto \mathbf{1}$, so after a simple normalization the trace is preserved. However, if $E_\alpha$ contains contractions then its square might involve complicated operators and there is no simple argument why the trace will be preserved.

In our case, if we treat the singlet states as a code subspace, it means that $E_\alpha^\dagger E_\beta$ should act as a scalar $N_{\alpha\beta}$ on singlet states. The product $E_\alpha^\dagger E_\beta$ can be decomposed into irreducible representations under the corresponding gauge group.

Now the theorem can be reformulated as:*errors are correctable if and only if the singlet channel consists of identity operators $\mathbf{1}$ and gauge group Casimirs (as they annihilate singlets).*

One last comment is that we will consider the case of erasures at known locations. It means that operator indices $I_l$ in $\alpha$ can be drawn from one fixed set of indices.

---

[12]For matrix models like BFSS $I = (a, i, j)$, where $a$ is $SO(9)$ spinor index and $ij$ are $SU(N)$ adjoint indices. For CTKT $I = (a, b, c)$, with $a, b, c$ being $O(N)^3$ fundamental indices.

## 3.2 Erasure and approximate error correction

In this subsection we will connect the discussion about the error operations in the previous subsection with holographic error correction described in Section 2. Following [3], we will argue that the ability to correct any error defined by a product of Pauli matrices leads to the ability to correct erasures of known fermions. We will also discuss this in the setting of approximate error correction.

How do we describe erasure of a subsystem $E$ with quantum operation (3.3)? One option is to use the so-called *depolarization channel*[13] which makes the reduced density matrix on $E$ maximally mixed:[14]

$$\mathcal{E}: \; \rho_{E\overline{E}} \to \widetilde{\rho}_{E\overline{E}} = \frac{1}{\dim_E} \mathbf{1}_E \otimes \rho_{\overline{E}}, \tag{3.12}$$

$$\rho_{\overline{E}} = \mathop{\mathrm{Tr}}_E \rho_{E\overline{E}}. \tag{3.13}$$

In th above equations we split the system into $E$ and the compliment of $E$ which we call $\overline{E}$. $\rho_{E\overline{E}}$ is the original density matrix:

$$\rho_{E\overline{E}} = \rho. \tag{3.14}$$

If $E$ is a one qubit, then $\mathcal{E}$ is given by:

$$\mathcal{E}: \; \rho \to \frac{1}{4}\mathbf{1}\rho\mathbf{1} + \frac{1}{4}X\rho X + \frac{1}{4}Y\rho Y + \frac{1}{4}Z\rho Z, \tag{3.15}$$

where Pauli matrices $\mathbf{1}, X, Y, Z$ act on $E$. Obviously, if $E$ consists of more than one qubit, we can depolarize all the qubits in $E$ one by one. Such operation will involve all possible Pauli strings supported on $E$. It was proven in [3] that this is equivalent to correcting the erasure of $E$. This is why we dedicated a lot of time to Majorana operator strings in the previous section.

Suppose now depolarization (3.12) is only approximately correctable. Meaning that there is recovery operation $\widetilde{\mathcal{R}}$:

$$\widetilde{\mathcal{R}}: \; \widetilde{\rho}_{E\overline{E}} \to \hat{\rho}_{E\overline{E}}, \tag{3.16}$$

such that $\hat{\rho}$ is close to the original $\rho$.

What do we mean by "close"? One measure is the conventional $L_2$ matrix norm, often called the trace distance:

$$D(\rho, \sigma) = ||\rho - \sigma|| = \frac{1}{2} \mathop{\mathrm{Tr}} \sqrt{(\rho - \sigma)^\dagger (\rho - \sigma)}. \tag{3.17}$$

Another important measure is fidelity. Fidelity $F$ between two density matrices are defined by:[15]

$$F(\rho, \sigma) = F(\sigma, \rho) = \mathop{\mathrm{Tr}} \sqrt{\rho^{1/2} \sigma \rho^{1/2}}. \tag{3.18}$$

In fact, these two are closely related. One can show that

$$1 - F \leq D \leq \sqrt{1 - F^2}. \tag{3.19}$$

So fidelity and trace distance are equivalent. In this paper we will mostly concentrate on fidelity.

---

[13]In quantum information theory "erasure channel" means a different thing [53]: it replaces a qubit by an erasure state $|2\rangle$ which is orthogonal to both spin-up $|1\rangle$ and spin-down $|0\rangle$.

[14]Usually people consider depolarization with probability $p$:

$$\rho_{E\overline{E}} \to (1-p)\rho_{E\overline{E}} + p \frac{1}{\dim_E} \mathbf{1}_E \otimes \rho_{\overline{E}},$$

but here we want to enforce the depolarization, thus $p = 1$.

[15]Uhlmann's theorem states that

$$F(\rho, \sigma) = \max_{|\xi\rangle, |\zeta\rangle} |\langle \xi | \zeta \rangle|,$$

where the maximum is over all possible purifications $|\xi\rangle, |\zeta\rangle$ of $\rho$ and $\sigma$.

# 4 Matrix models

In this Section we will discuss error correcting properties of fermionic singlet sector in various matrix quantum mechanical models. Our discussion will not be tied to a particular Hamiltonian.

This Section is organized as follows. In Section 4.1 we describe the field content of matrix models we study and how we build error operators. In Section 4.2 we discuss KL condition and how it is related to operator spectrum. Also in this section we describe our main observation, which is the suppression of singlets in KL condition. In Section 4.3 we formulate our main result. In Section 4.4 we explain how error correction breaks for large amount of erasures.

## 4.1 Setup and error operators

We will consider a collection of Majorana fermions in the adjoint representation of $SU(N)$:

$$\psi^a_{ij}, \tag{4.1}$$

where $a = 1, \ldots, D$ is "flavor" index and $ij$ are matrix indices of $SU(N)$ adjoint matrix, which is traceless hermitian matrix. For BFSS $D = 16$. We may want to define error operators in terms of $\psi^a_{ij}$ strings:

$$E_\alpha \propto \psi^a_{ij}\psi^b_{kl}\ldots \tag{4.2}$$

However, this is a bad choice because $E^\dagger_\alpha E_\alpha$ is not proportional to identity operator.[16] In other words, $\psi^a_{ij}$ are not Pauli strings. Moreover they are not independent.[17] What we can do is to introduce a proper orthogonal basis $T^{(ij)}$ in the space of $SU(N)$ algebra adjoints:

$$\text{Tr}\left(T^{(ij)}T^{(kl)}\right) = \delta^{(ij),(kl)}. \tag{4.3}$$

Variable $(ij)$ should be understood as a single variable taking $\dim su(N) = N^2 - 1$ values. Now

$$\psi^a_{ij} = \sum_{(kl)} T^{(kl)}_{ij}\psi^a_{(kl)} \tag{4.4}$$

and $\psi^a_{(ij)}$ are hermitian operators $\left(\psi^a_{(ij)}\right)^\dagger = \psi^a_{(ij)}$ which square to one:

$$\{\psi^a_{(ij)}, \psi^b_{(kl)}\} = 2\delta^{ab}\delta_{(ij),(kl)}. \tag{4.5}$$

We focus on erasures of fermions at $S_0$ known locations. As we have explained in the previous section it means that the error operators $E_\alpha$ are given by products of $\psi^a_{ij}$:

$$E_\alpha = \frac{1}{\sqrt{M_0}} \prod_{(a_l,(ij)_l)\in\alpha} \psi^{a_l}_{(ij)_l}, \tag{4.6}$$

where index $l$ enumerates pairs $I = (a, (ij))$ in the string $\alpha$. Having erasures at known locations means that pairs $I$ are to be drawn from a particular fixed set of $S_0$ triples. This way $M_0 = 2^{S_0}$.

---

[16]$\psi^a_{ij}$ have anti-commutation relations of Clifford algebra:

$$\{\psi^a_{ij}, \psi^b_{kl}\} = 2\delta^{ab}\left(\delta_{il}\delta_{jk} - \frac{1}{N}\delta_{ij}\delta_{kl}\right)\mathbf{1}.$$

The $ijkl$ color structure in the right hand side is specific to $SU(N)$. $1/N$ term is needed to make it traceless in $ij$ and $kl$. Also $\psi^a_{ij}$ are hermitian, but not as operators, but in the matrix sense:

$$\left(\psi^a_{ij}\right)^\dagger = \psi^a_{ji}.$$

[17]Because of the traceless condition: $\sum_i \psi^a_{ii} = 0$.

## 4.2 Operator spectrum

According to KL theorem we are interested in projecting the product $E_\alpha^\dagger E_\beta$ onto singlet(code) subspace:

$$P_{\text{code}} E_\alpha^\dagger E_\beta P_{\text{code}}. \tag{4.7}$$

If $\alpha = \beta$ then due to Majorana relation (4.5) we have only identity operator: $E_\alpha^\dagger E_\alpha \propto \mathbf{1}$.

More generally, for $\alpha \neq \beta$, we will have a bunch of non-trivial singlet operators which we collectively call $\mathcal{O}_{\alpha\beta}$:

$$P_{\text{code}} E_\alpha^\dagger E_\beta P_{\text{code}} = \frac{1}{M_0} \left( \delta_{\alpha\beta} P_{\text{code}} + \mathcal{O}_{\alpha\beta} P_{\text{code}} \right). \tag{4.8}$$

*Our main observation is that matrix elements of $\mathcal{O}_{\alpha\beta}$ in singlet states are small.*

Let us see what singlet operators the product $E_\alpha^\dagger E_\beta$ may have. The fermions have peculiar $(ij)$ indices, which label orthogonal $su(N)$ algebra generators. How do we contract them to form a singlet operator?

The inverse of eq. (4.4) is

$$\psi^a_{(kl)} = \text{Tr}\left( T^{(kl)} \psi^a \right) = \sum_{ij} T^{(kl)}_{ij} \psi^a_{ji}. \tag{4.9}$$

After that we can easily form a singlet with matrices $\psi^a_{ji}$. The matrices $T^{(ij)}$ will automatically form the same singlet. For example:

$$\psi^1_{(ij)} \psi^2_{(kl)} = \frac{1}{N^2 - 1} \text{Tr}\left( \psi^1 \psi^2 \right) \text{Tr}\left( T^{(ij)} T^{(kl)} \right) + [\text{non-singlets}], \tag{4.10}$$

$$\psi^1_{(ij)} \psi^2_{(kl)} \psi^3_{(pq)} = \frac{1}{N^3 - 1/N} \text{Tr}\left( T^{(ij)} T^{(kl)} T^{(pq)} \right) \left( \text{Tr}\left( \psi^1 \psi^2 \psi^3 \right) + \frac{1}{N^2} \text{Tr}\left( \psi^1 \psi^3 \psi^2 \right) \right) -$$
$$\frac{1}{N^3 - 1/N} \text{Tr}\left( T^{(ij)} T^{(pq)} T^{(kl)} \right) \left( \frac{1}{N^2} \text{Tr}\left( \psi^1 \psi^2 \psi^3 \right) + \text{Tr}\left( \psi^1 \psi^2 \psi^3 \right) \right) + [\text{non-singlets}]. \tag{4.11}$$

In the large $N$ limit we can neglect the difference between $N^3 - 1/N$ and $N^3$. Also in the above example we had a mixing between $\text{Tr}\left( \psi^1 \psi^2 \psi^3 \right)$ and $\text{Tr}\left( \psi^1 \psi^3 \psi^2 \right)$.[18] Obviously if $N$ is large and the number of operators is not too big, such "wrong contractions" will be suppressed. We will comment on this more in the next Section. So we simply write

$$\psi^1_{(ij)} \psi^2_{(kl)} \psi^3_{(pq)} = \frac{1}{N^3} \text{Tr}\left( T^{(ij)} T^{(kl)} T^{(pq)} \right) \text{Tr}\left( \psi^1 \psi^2 \psi^3 \right) - \frac{1}{N^3} \text{Tr}\left( T^{(ij)} T^{(pq)} T^{(kl)} \right) \text{Tr}\left( \psi^1 \psi^2 \psi^3 \right). \tag{4.12}$$

As we have mentioned in the Introduction, we can bound various fermionic operators by considering a representation theory of auxiliary $SU(N)$ groups. In particular, consider $SU(N)_a$ which rotates fermion $\psi^a_{ij}$ only. Singlet under the original gauge $SU(N)$ are not necessarily singlets under $SU(N)_a$. But since we are dealing with fermions, we would not have arbitrary "big" representations. Therefore the Casimir will be bounded. In Appendix B we show that the quadratic Casimir $C_2(SU(N)_a)$ of $SU(N)_a$ is proportional to $\text{Tr}(\psi^a)^4$. Whereas this operator can be bounded by $2N^3$:

$$C_2(SU(N)_a) \propto \text{Tr}\left( \psi^1 \psi^1 \psi^1 \psi^1 \right) = \left( N^2 - 1 \right) \left( 2N - \frac{1}{N} \right) \times \mathbf{1} \leq 2N^3 \times \mathbf{1}. \tag{4.13}$$

---

[18]Since $\psi^1_{ij} \psi^2_{kl} \psi^3_{pq} \propto \delta_{jk} \delta_{lp} \delta_{qi} \text{Tr}\left( \psi^1 \psi^2 \psi^3 \right) - \delta_{jp} \delta_{qk} \delta_{li} \text{Tr}\left( \psi^1 \psi^3 \psi^2 \right)$, it happened because the two delta symbols can be both non-zero for some small amount of indices.

Notice that naively we expect that this operator scales as $N^4$.

By using various tricks with cutting and gluing $\psi$ we can bound any fermionic operator by an appropriate combination of Casimirs. The only exception are bilinears like $\text{Tr}\left(\psi^1\psi^2\right)$. One can easily check that for them the naive expectation $N^2$ is true:

$$\text{Tr}\left(\psi^a\psi^b\right) \propto N^2. \tag{4.14}$$

We will make sure we avoid them. Essentially it means that the set of erasures/errors can contain each $SU(N)$ adjoint index $(ij)$ only once, because bilinears come with a color factor $\text{Tr}\left(T^{(ij)}T^{(kl)}\right) = \delta^{(ij),(kl)}$.

One final comment is that color factors like $\text{Tr}\left(T^{(ij)}T^{(kl)}\ldots\right)$ are not large. In Section B.3 we demonstrate that they can be bounded by $\sqrt{2}$.

In Appendix B we prove the following bound:

**Theorem 2.** *For any single-trace fermionic operator $\mathcal{O}_k$ made from $3 \leq k \leq 2\sqrt{N}$ fermions, there is a bound on the matrix elements in singlet states:*

$$|\mathcal{O}_{3m}| \leq 2^m N^{5m/2},$$
$$|\mathcal{O}_{3m+1}| \leq \sqrt{2}N \times 2^m N^{5m/2},$$
$$|\mathcal{O}_{3m+2}| \leq 2N^2 \times 2^m N^{5m/2}, \tag{4.15}$$
$$m \geq 1,$$

*and*

$$|\mathcal{O}_4| \leq 2N^3, \tag{4.16}$$

*where the $|\cdot|$ mean the element-wise matrix norm in the singlet subspace:*

$$|\mathcal{O}| = \max_{|s_{1,2}\rangle}|\langle s_1|\mathcal{O}|s_2\rangle|, \quad \langle s_i|s_i\rangle = 1. \tag{4.17}$$

In terms of fermionic number $k$ the above bounds can be concisely formulated for any operators, not necessarily single-trace:

$$|\langle s_1|\mathcal{O}_k|s_2\rangle| \leq 2^k N^{k-k/10}, \quad k \geq 3, \tag{4.18}$$

as long as $\mathcal{O}_k$ does not contain bilinear operators, such as $\text{Tr}\left(\psi^1\psi^2\right)$, since they are of order $N^2$, and therefore are not suppressed compared to naive $N$ counting.

## 4.3 Main result

Let us return to the error correction and formulate our main theorem. One has to be careful about translating $1/N$ correction in (4.8) to the actual fidelity of recovery $\widetilde{R}$. Appendix A is dedicated to this question.

Finally, we state our main result:

**Theorem 3.** *(Error correction in matrix models) Gauge-singlet sector of matrix models forms an approximate error correcting code against any erasure of $S_0$ fermions at known locations as long as*

$$S_0 \leq N^{1/10} \tag{4.19}$$

*and the set of erased fermions does not contain one $SU(N)$ index $(ij)$ more than once.*

*Recovery fidelity can be bounded by:*

$$F \geq 1 - C_{\text{MM}}\frac{S_0}{N^{2/5}}, \quad C_{\text{MM}} = 20736. \tag{4.20}$$

This theorem is proved in Appendix A. $C_{\text{MM}}$ is a product of three different factors and the fact that it is huge stems from using loose upper bounds for all of them.

## 4.4 Large operators

As we mentioned in the Introduction, $\mathcal{O}_{\alpha\beta}P_{\text{code}}$ might not be suppressed for complicated operators.

This happens because $\mathcal{O}_{\alpha\beta}$ might contain a lot of singlets. Let us explain how it might happen. This will give us an estimate of how many erasures we can correct. Considerations below are very crude, so they could probably be refined. Let us consider $\mathcal{O}_{\alpha\beta}$ build from $k$ fermions. We need to estimate its matrix elements in singlet states.

1. First of all, in the previous Section we neglected operators with "wrong contractions". Thanks to Theorem 2 they have the same scaling with $N$. So a problem might occur only if they are too many. In Appendix B.4 we show is indeed not a problem as long as $k \leq N/2$. For such $k$ they introduce an extra factor of $3/2$ at worst.

2. In Appendix B.3 we show that color factors $\text{Tr}\left(T^{(ij)}T^{pq}\dots\right)$ are bounded by $\sqrt{2}$ regardless of $k$.

3. Product $E_{\alpha}^{\dagger}E_{\beta}$ of length $k$ typically contain several singlet operators. Expectation value of each of them is suppressed by $2^k/(N^{k/10})$ due to eq. (4.18).

4. The number of different singlet operators build from $k$ operators $\psi_{ij}^a$ can be generously bounded[19] by the number of singlet representations in the tensor product of $k$ adjoint representations of $SU(N)$. This can be computed by the following integral over unitary matrices:

$$\int_{SU(N)}[dU]\left(\text{Tr}\,U\,\text{Tr}\,U^{\dagger}-1\right)^k, \tag{4.21}$$

where $\text{Tr}\,U\,\text{Tr}\,U^{\dagger}-1$ is the character of the adjoint representation. Using the following large $N$ result [54]

$$\int_{SU(N)}[dU]\prod_l\left(\text{Tr}\,U^l\right)^{a_l}\text{Tr}\left(U^{l,\dagger}\right)^{b_l}=\prod_l\delta_{a_l,b_l}l^{a_l}(a_l)! \tag{4.22}$$

We see that this number is bounded[20] by $k!$.

We conclude that for $\alpha\beta$ of length $k$, the singlet contribution is bounded by

$$\frac{3}{\sqrt{2}}2^k\frac{k!}{N^{k/10}}. \tag{4.23}$$

It is clear that this number is small as long as errors $E_{\alpha}$ include no more than

$$\frac{N^{1/10}}{2} \tag{4.24}$$

fermions.

Unfortunately, for our proof presented in Appendix A, it is not enough that matrix elements of $\mathcal{O}_{\alpha\beta}P_{\text{code}}$ are small. We will need to show that sums like

$$\sum_{s_1,\alpha:\alpha\neq\beta}\langle s_1|\mathcal{O}_{\alpha\beta}|s_2\rangle^l=\sum_{s_1,\alpha:\alpha\neq\beta}\langle s_1|E_{\alpha}^{\dagger}E_{\beta}|s_2\rangle^l \tag{4.25}$$

are small, where $|s_{1,2}\rangle$ are singlet states. Notice, that this can not be small for errors at unknown locations. For example, $\beta$ can be empty string. Then even if $\alpha$ includes only two fermions, the sum over $\alpha$ will have $N^2$ terms, overcoming $1/N$ suppression.

---

[19]It is an upper bound because we do not have bilinears and we do not take into account fermionic symmetry. However for large $k$ taking into account these effect will only modify the numerical prefactor. So we do not overcount too much.

[20]Note that this number includes non-single trace operators. Large $N$ matrix theories have Hagedorn transition [41] characterized by *exponential* growth of the number of *single-trace* operators.

# 5 Tensor models

In this Section we analyze error correcting properties of fermionic tensor models. Our results and the general discussion are very similar to the Section 4 about matrix models.

This Section is organized as follows. In Section 5.1 we describe the field content of tensor models and state various elementary properties of CTKT model. In Section 5.2 we formulate our main result. In Section 5.3 we discuss KL condition and operator spectrum. In Section 5.4 we explain how error correction breaks for large amount of erasures. Finally, in Section 5.5 we will discuss the correction of less general errors which are specific to tensor models.

## 5.1 Setup of the model

CTKT model is quantum mechanical model including $N^3$ Majorana fermions $\psi_{abc}$, $a, b, c = 1, \dots, N$. Fermions can be rotated with large symmetry $O(N)^3$ group. More generically, one can consider $O(N_1) \times O(N_2) \times O(N_3)$ model with Majorana fermion anti-commutation relation:

$$\{\psi_{a_1 b_1 c_1}, \psi_{a_2 b_2 c_2}\} = 2\delta_{a_1 a_2} \delta_{b_1 b_2} \delta_{c_1 c_2} \mathbf{1}. \tag{5.1}$$

Unlike matrix model case, they are truly hermitian:

$$(\psi_{abc})^\dagger = \psi_{abc}. \tag{5.2}$$

This model has very large symmetry group $G = O(N_1) \times O(N_2) \times O(N_3)$. Corresponding generators $Q^1, Q^2, Q^3$ read as

$$Q^1_{aa'} = \frac{i}{2} \sum_{bc} [\psi_{abc}, \psi_{a'bc}],$$

$$Q^2_{bb'} = \frac{i}{2} \sum_{ac} [\psi_{ab'c}, \psi_{ab'c}],$$

$$Q^3_{cc'} = \frac{i}{2} \sum_{ab} [\psi_{abc}, \psi_{abc'}]. \tag{5.3}$$

And the corresponding quadratic Casimir operators $C_2$ are

$$C^1_2 = \sum_{a \neq a'} Q^1_{aa'} Q^1_{a'a},$$

$$C^2_2 = \sum_{b \neq b'} Q^2_{bb'} Q^2_{b'b},$$

$$C^3_2 = \sum_{c \neq c'} Q^3_{cc'} Q^3_{c'c}. \tag{5.4}$$

As was mentioned in the Introduction, the number of singlet states is exponentially big:

$$\text{Singlet states} = 2^{N^3/2 - cN^2 \log N}, \quad c = \text{const.} \tag{5.5}$$

Also there are a lot of singlet operators. The number of singlet operators build from $2k$ fermions grows factorially [42]:

$$\#[\mathcal{O}_{2k}] \sim 2^k k!. \tag{5.6}$$

In principle, for singlet states one can exclude operators containing $O(N)$ charge operators 5.3. However, the above asymptotic formula still holds even for this subset.

Although we would not assume any dynamical information, it is still worth recalling some properties of the CTKT Hamiltonian.

The CTKT Hamiltonian has the following explicit expression:

$$H_{\text{CTKT}} = J \sum_{abca'b'c'} \psi_{abc}\psi_{a'b'c}\psi_{a'bc'}\psi_{ab'c'}. \tag{5.7}$$

In the large $N$ limit this model is also dominated by melonic diagrams, leading to SYK $q = 4$ $G\Sigma$ action. One can think about the CTKT Hamiltonian as $N_{\text{SYK}} = N_1 N_2 N_3$ SYK model with very sparse $J_{ijkl}$. Namely, there are only $N_{\text{SYK}}^2$ non-zero terms out of $N_{\text{SYK}}^4$ possible.

## 5.2 Main result

As in the case of matrix models we need to study the matrix elements of product of fermionic operators. We focus on erasures at known locations, such that error operators $E_\alpha$ are given by products of $\psi_{abc}$:

$$E_\alpha = \frac{1}{\sqrt{M_0}} \prod_{(a_l,b_l,c_l) \in \alpha} \psi_{a_l b_l c_l}, \tag{5.8}$$

where index $l$ enumerates triples $I = (a, b, c)$ in the string $\alpha$. Having erasures at known locations means that triples in $\alpha$ are to be drawn from a particular fixed set of $S_0$ triples. This way $M_0 = 2^{S_0}$.

Similarly to matrix-model case, first we will need a bound on matrix elements of singlet operators. In Appendix C we prove the following bound:

**Theorem 4.** *For any singlet state $|s\rangle$ in tensor quantum mechanics and any singlet operator build from $2k$ fermions $\psi_{abc}$:*

$$|\langle s|\mathcal{O}_{2k}|s\rangle| \leq N^{5k/2} \times \begin{cases} 1, k \text{ even} \\ 1/\sqrt{N}, k \text{ odd} \end{cases}, \tag{5.9}$$

*as long as $k \leq \sqrt{N}$.*

Similarly to the matrix model case, this bound is based on simple properties of $O(N)$ Casimirs. First we bound Casimir operators by $N^5$ and then we use cutting and gluing to bound all other operators by powers of Casimirs.

Now we can formulate the statement about the error correction:

**Theorem 5.** *(Error correction in CTKT) All singlet states in CTKT model form an approximate quantum error correcting code against errors $\{E_\alpha\}$ at $S_0$ known locations as long as*

$$S_0 \leq \frac{N^{1/6}}{2}. \tag{5.10}$$

*Recovery fidelity can be bounded by*

$$F \geq 1 - C_{\text{CTKT}} \frac{S_0^5}{N^2}, \quad C_{\text{CTKT}} = 67584. \tag{5.11}$$

This theorem is proved in Appendix A. Like $C_{\text{MM}}$, $C_{\text{CTKT}}$ is a product of three different factors and that fact that it is huge stems from using loose upper bounds for all of them.

## 5.3 Operator spectrum

As in the case of matrix models, the previous theorem uses various facts about $N$ scaling of singlet operators.

Let us discuss the behavior of matrix elements of $E_\alpha^\dagger E_\beta$ in singlet states. For example, if operators $E_\alpha$ include only single fermion operators, then $\langle s_1 | E_\alpha^\dagger E_\beta | s_2 \rangle$ will vanish unless $\alpha = \beta$, since there are no non-trivial singlets formed from two $\psi$ operators:

$$\psi_{abc} \psi_{a'b'c'} \sim \mathbf{1} \times \delta_{aa'} \delta_{bb'} \delta_{cc'} + \text{non-singlets such as } Q^1_{aa'}. \tag{5.12}$$

However, if $E_\alpha$ are two-fermion operators then their product[21] might contain the Hamiltonian and Casimir operators 5.4:

$$\begin{aligned}
\psi_{a_1 b_1 c_1} \psi_{a_2 b_2 c_2} \psi_{a_3 b_3 c_3} \psi_{a_4 b_4 c_4} \sim \mathbf{1} \times & \left( \delta^{a_1 b_1 c_1}_{a_2 b_2 c_2} \delta^{a_3 b_3 c_3}_{a_4 b_4 c_4} + \delta^{a_2 b_2 c_2}_{a_3 b_3 c_3} \delta^{a_1 b_1 c_1}_{a_4 b_4 c_4} - \delta^{a_1 b_1 c_1}_{a_3 b_3 c_3} \delta^{a_2 b_2 c_2}_{a_4 b_4 c_4} \right) \\
+ & \frac{H_{\text{CTKT}}}{N^6} \times \left( \delta^{a_1}_{a_2} \delta^{a_3}_{a_4} \delta^{b_1}_{b_3} \delta^{c_1}_{c_4} \delta^{b_2}_{b_4} \delta^{c_2}_{c_3} + [\text{5 other combinations}] \right) \\
+ & \frac{C^1_2}{N^6} \times \left( \delta^{b_1 c_1}_{b_2 c_2} \delta^{b_3 c_3}_{b_4 c_4} \delta^{a_1 a_2}_{a_3 a_4} + [\text{2 other combinations}] \right) \\
+ & [\, C^2_2, C^3_2 \text{ Casimirs }\,] + [\text{non-singlets}]. \tag{5.13}
\end{aligned}$$

The numerical factor in the above expression have $1/N$ corrections and tensor $\delta$ with multiple indices is defined as

$$\delta^{i_1 \dots i_n}_{j_1 \dots j_n} = \prod_{l=1}^{n} \delta^{i_l}_{j_l}. \tag{5.14}$$

Casimir operators act by zero on the singlet sector, so their presence does not cause any problems. However, the Hamiltonian does not act as identity. Moreover, if we increase the number of $\psi$, there is a factorially growing number of singlet operators [42].

Actually, we do not need to get rid of all the singlets. Casimir operators and identity operators are good, so we can keep them. Notice, that all operators come with certain $\delta^a_{a'} \dots$ structure. One obvious thing to do is to make sure this structure is zero in front of bad singlet operators. This requires restricting the set of $a, b, c$ indices in $E_\alpha$. By doing this we allow $E_\alpha$ to have length $\sim N$, but restricting the set of color indices. We will present the details in Subsection 5.5

Now we are going to bound the singlet contribution to $E_\alpha^\dagger E_\beta$ by $1/N$ as long as $E_\alpha$ are not very long. The key element is the bound on matrix element we mentioned earlier.

## 5.4 Large operators

Again parallel to matrix model case, for large $k$ there could be too many different singlets in $E_\alpha^\dagger E_\beta$ and they can overcome $1/N$ suppression.

Now we need to compute the number of singlets arising in the product of $2k$ 3-tensor fermionic fields. This will include both different singlet operators and different delta-function color factors. Like in the matrix model case, we neglect the fermionic symmetry and obtain the following upper bound:

$$\left( \int_{SO(N)} [dM] (\text{Tr} M)^{2k} \right)^3. \tag{5.15}$$

---

[21]In the below we are assuming large $N$ limit and neglect what we called "wrong contractions" in matrix model. Essentially the problem is that delta-functions below are not orthogonal. Using the reasoning similar to Appendix B.4 one can show that it is not important as long as $k \leq N/2$.

Using the following large $N$ result [54]

$$\int_{SO(N)} [dM] (\operatorname{Tr} M)^{2k} = 2^k \frac{1}{\sqrt{\pi}} \Gamma\left(k + \frac{1}{2}\right), \tag{5.16}$$

we get that the number is bounded by

$$\left(2^k k!\right)^3. \tag{5.17}$$

Finally, using the Stirling approximation we see that the total singlet contribution in the product $E_\alpha^\dagger E_\beta$ can be bounded by:

$$[E_\alpha^\dagger E_\beta]_{\text{singlet}} \sim \frac{1}{N^{3k}} \times N^{5k/2} \times 2^{3k} k!^3 \sim \left(\frac{8k^3}{\sqrt{N}}\right)^k. \tag{5.18}$$

So that this contribution is small for $k \lesssim N^{1/6}/2$.

## 5.5 Exact error correction for less general errors

Now we consider possible restrictions on $E_\alpha$ to get rid of "bad" singlet operators. In deriving the expansion (5.13) we essentially used Wick theorem. Notice, if we have two $\psi$-operators, $\psi_{abc}$ and $\psi_{a'b'c'}$, contracting just one index, for example $a-a'$, leads to a non-singlet operator $\psi_{abc}\psi_{ab'c'}$ from which one can build either a charge operator or a more complicated (bad) singlet. Charge operators(and identity) are characterized by the property that once we contact one pair of indices($a$ in this case) for two particular $\psi$, at least one other pair ($b$ or $c$) will be contracted between them too.

This observation leads to the following naive solution. We restrict the set of $\psi$ in $E_\alpha$ such that contraction of one index in a pair of $\psi$ will lead to the contraction of another pair. This can be done as follows. Suppose that instead of generic $O(N_1) \times O(N_2) \times O(N_3)$ model we have $O(N)^3$ model. Then we restrict the set of $\psi_{abc}$ in $E_\alpha$, from $N^3$ to $N^2$ by selecting two indices, for example $a$ and $b$, and a permutation $\sigma$ of $N$ elements, such that the only allowed $\psi$ are $\psi_{a\ \sigma(a)\ c}$. However, it is not enough to ensure that contraction in $a$ will lead to a contraction in $b$, since there might be several $\psi$ with the same $a$. So we further require that all $a$ are different. We have essentially proven (by construction) the following theorem.

**Theorem 6.** *Pick up a permutation $\sigma$ of $N$ elements. Suppose that the error operators $\{\widetilde{E}_\alpha\}$ contain fermionic strings*

$$\prod_{(a_l, b_l, c_l) \in \alpha} \psi_{a_l b_l c_l} \tag{5.19}$$

*such that*

- *$b_l = \sigma(a_l)$, $l = 1, \ldots, d$.*

- *All $a_l$ are different.*

*The number of such operators is $\frac{N^{2d}}{d!}\left(1 + O(ld/N)\right)$. Then quantum operations build from $\widetilde{E}_\alpha$ are correctable.*

Obviously, such $\widetilde{E}_\alpha$ can not include more than $N$ different fermions.

# 6 Conclusion

In this paper we investigated error correcting properties of the singlet subsector of matrix and tensor models. Our results are purely "kinematical": they rely on large $N$ limit only, and do not involve any dynamical information. For example, they are applicable to the well-studied BFSS matrix model and CTKT tensor quantum mechanics. This provides further evidence that AdS/CFT is tightly related to QEC. For simplicity we concentrated on fermionic subsector which is finite dimensional. We have found that singlets indeed can correct for erasures at known locations provided that there are not too many of them.

As was expected from holographic picture, the error correction is exact only at infinite $N$ limit. In general, we have managed to bound the recovery fidelity by $1/N^{\#}$ (i.e. $1/N$ in some fractional power).

Unfortunately, the models we studied do not have any spatial structure, so it is very difficult to probe the bulk in them. Moreover, for generic matrix/tensor models the bulk is highly stringy. BFSS does have a well-defined ten-dimension bulk, but it is still not clear what is the boundary counterpart of extremal surfaces in the bulk [55].

Obviously it would be very interesting to study QEC in systems with spatial structure. One possibility would be to study coupled SYK models or coupled tensor models, since at low temperatures they are dual to traversable wormholes [56] and in this case one can even talk about RT surfaces. The mass deformation of BFSS model, known as BMN possesses M2/M5 brane vacua which are realized as fuzzy-spheres. It would be interesting so study the error correction in this case and involve some dynamical input. It is expected that low-energy states of BMN/BFSS have 't Hooft scaling (D.4) in the matrix elements. As was noted in the main text, 't Hooft scaling grows slower with $N$, than the bounds we used, so assuming 't Hooft scaling would yield better bounds.

It might seem that our results are quite weak: we can correct errors involving only $1/N$ fraction of fermions at best. However, in our setup we can reconstruct the original state *completely*. Recall the Figure 1 which was the starting point for holographic error correction. In this setup erasure of $A, B, C$ should not affect point $O$. However, upon erasure of $A$, for example, the bulk information inside its casual wedge $\mathcal{W}_C[A]$ is lost. So from the boundary perspective we should not expect to reconstruct the original state after big erasures, since some information is really lost. It was proposed in [3] that this should correspond to the so-called *subsystem quantum error correction*. It would be interesting to study this setup for singlets and see how much the space of errors can be enlarged.

# Acknowledgment

I am indebted to J. Maldacena for numerous comments and discussions. I am grateful to I. Danilenko for answering numerous questions about Lie algebras, A. Alfieri for pointing out orthogonal Procrustes problem and C. King for helping to find the proper cuts for 10-particle operators in tensor models. Also I would like to thank A. Almheiri, X. Dong, A. Dymarsky, A. Gorsky, I. Klebanov, D. Marolf, K. Pakrouski, F. Popov and B. Swingle for discussions and comments on the manuscript.

# A  Proving the bound on recovery fidelity

## A.1  Notation

Let us fix the notation:

- Big Latin indices, $I, J, \ldots$ denote indices of Majorana fermions.

- Small Latin indices from the beginning of the alphabet, $a, b, c, \ldots$ denote the orthonormal basis in the code subspace. For example, the projector $P_{\text{code}}$ can be written as:

$$P_{\text{code}} = \sum_c |c\rangle\langle c|. \tag{A.1}$$

- Greek indices $\alpha, \beta$ denote a string of $\psi_{I_1} \psi_{I_2} \ldots$ of length no more than $S_0$, with indices drawn from a particular subset of $S_0$ elements.

- We consider the error operators $E_\alpha, \alpha = 1, \ldots, M_0$ acting on density matrices as

$$\rho \to \sum_\alpha E_\alpha \rho E_\alpha^\dagger. \tag{A.2}$$

- Their normalization:

$$E_\alpha^\dagger E_\alpha = \frac{1}{M_0} \quad \text{(no sum)}. \tag{A.3}$$

- Up to a normalization, $E_\alpha$ is a unitary operator. We denote to corresponding unitary by

$$U_\alpha = \sqrt{M_0} E_\alpha. \tag{A.4}$$

- Operators $E_\alpha$ satisfy the following analogue of Knill–Laflamme condition:

$$P_{\text{code}} E_\alpha^\dagger E_\beta P_{\text{code}} = \frac{1}{M_0} \left( \delta_{\alpha\beta} + \mathcal{O}_{\alpha\beta} \right) P_{\text{code}}, \tag{A.5}$$

where $\mathcal{O}_{ij}$ commutes with $P_{\text{code}}$. As follows from this requirement: $\mathcal{O}_{\alpha\alpha} = 0$.

- Finally, states $|c\ \alpha\rangle$ denote normalized states:

$$|c\ \alpha\rangle = U_\alpha |c\rangle. \tag{A.6}$$

These states are charged and do not belong a code subspace. Unfortunately, these states are not orthogonal. However, they are almost orthogonal.

$$\langle a\ \alpha | b\ \beta \rangle \propto \frac{1}{N} \neq 0, \ \alpha \neq \beta. \tag{A.7}$$

We will make this statement precise later.

## A.2  Idea of the proof

The proof is inspired by the proof of Theorem 10.1 in [44]. The main challenge is the fact that naively defined syndrome operators do not form a well-defined quantum operation. Let us define naive projectors $P_\alpha$:

$$P_\alpha = U_\alpha P_{\text{code}} U_\alpha^\dagger = \sum_c |c\ \alpha\rangle\langle c\ \alpha|. \tag{A.8}$$

And naive recovery $\mathcal{R}$:

$$\mathcal{R}(\sigma) = \sum_{\alpha} U_{\alpha}^{\dagger} P_{\alpha} \sigma P_{\alpha} U_{\alpha} + P_R \sigma P_R. \tag{A.9}$$

We will need to find projector $P_R$ which completes the set of projectors $P_{\alpha}$. This recovery recovers well: in the sum over $\alpha$ and $\beta$:

$$\sum_{\alpha,\beta} U_{\alpha}^{\dagger} P_{\alpha} E_{\beta} \rho E_{\beta}^{\dagger} P_{\alpha} U_{\alpha}, \tag{A.10}$$

the term with $\alpha = \beta$ is exactly the original density matrix and the terms with $\alpha \neq \beta$ are suppressed by $1/N$. However, this is not a trace-preserving operation: trace preservation requires the operator $P_R$ to be:

$$\mathbf{1} - \sum_{\alpha} P_{\alpha} = P_R^{\dagger} P_R. \tag{A.11}$$

However, because of (A.7), different projectors $P_{\alpha}$ are not orthogonal. Therefore the left hand side is not even a positive operator. Despite the fact that $P_{\alpha}$ are almost orthogonal, we need them to be exactly orthogonal.

We will show in the next subsection (Section A.3) that there exists an orthogonalization $\widetilde{P_{\alpha}}$ of $P_{\alpha}$ which is close to $P_{\alpha}$. After that we will define an improved recovery $\widetilde{\mathcal{R}}$:

$$\widetilde{\mathcal{R}}(\sigma) = \sum_{\alpha} U_{\alpha}^{\dagger} \widetilde{P_{\alpha}} \sigma \widetilde{P_{\alpha}} U_{\alpha} + \widetilde{P_R} \sigma \widetilde{P_R}. \tag{A.12}$$

After obtaining simple bounds on distance between $P_{\alpha}$ and $\widetilde{P_{\alpha}}$ in Sections A.6.2 and A.7.2, it will be a simple algebraic exercise to show that $\widetilde{\mathcal{R}}$ has high fidelity.

## A.3 Orthogonal Procrustes problem

So we are facing the following geometric problem: we have a set of normalized vectors $|a\,\alpha\rangle$, which are almost orthogonal. We want to define a new set $|\widetilde{a\,\alpha}\rangle$ which are normalized and orthogonal. Obviously there are infinitely many choices of orthogonal basis $|\widetilde{a\,\alpha}\rangle$. We want the one which maximizes $|\langle a\,\alpha|\widetilde{a\,\alpha}\rangle|$.

This problem can be easily solved explicitly. In mathematical literature this is knows as orthogonal Procrustes problem. To simplify our notation a bit, let us define $v_i = |a\,\alpha\rangle$, so that small Latin indices from the middle of the alphabet denote the states: $i = \{a\,\alpha\}$. We want to find orthonormal basis $e_i$ closest to $v_i$. Let us denote the Gram matrix of vectors $v_i$ as $V_{ij}$:

$$V_{ij} = (v_i, v_j). \tag{A.13}$$

Matrix $V_{ij}$ is hermitian and positive-definite(since the scalar product is positive-definite). It can be diagonalized with the help of unitary $U$, where diagonal matrix $D$ has positive entries:

$$V = U^{\dagger} D U. \tag{A.14}$$

Denote the scalar products of $e_i$ and $v_j$ by $M_{ij}$:

$$M_{ij} = (e_i, v_j). \tag{A.15}$$

In general $M$ is not hermitian. But obviously we have the following matrix equation:

$$V = M^{\dagger} M. \tag{A.16}$$

General solution of this equation is

$$M = K\sqrt{D}U, \tag{A.17}$$

where $K$ is another unitary. We need to minimize the distance from $M$ to identity matrix:

$$\min \mathrm{Tr}(M-\mathbf{1})^\dagger (M-\mathbf{1}). \tag{A.18}$$

It is easy to see that the extremum of this problem is equivalent to extremizing the trace of $M+M^\dagger$. Recalling eq. (A.17), we see that the extremal $K = U^\dagger$, so that $M$ is a square-root of $V$:

$$V = M^2. \tag{A.19}$$

This answer make sense: if $V$ is close to identity, as we expect, $M$ is close to identity too. Namely, denote $\delta V = V - \mathbf{1}$. Then

$$M = \sqrt{\mathbf{1} + \delta V} = \mathbf{1} + \delta M = \mathbf{1} + \sum_{k=1}^{+\infty} C_{1/2}^k (\delta V)^k. \tag{A.20}$$

Returning to the previous notation, vectors $e_i$ are $|a\,\tilde{\alpha}\rangle$. So we see that

$$\langle \widetilde{a\,\alpha}|b\,\beta\rangle = M_{b\,\beta}^{\widetilde{a\,\alpha}} = \delta_{b\,\beta}^{a\,\alpha} + (\delta M)_{b\,\beta}^{a\,\alpha} = \delta_{b\,\beta}^{a\,\alpha} + \sum_{k=1}^{+\infty} C_{1/2}^k (\delta V^k)_{b\,\beta}^{a\,\alpha}. \tag{A.21}$$

It would be convenient to introduce the following bounds for the Gram matrix $\delta V$:

$$|[(\delta V)^k]_{b\,\beta}^{a\,\alpha}| \le \eta\,(\epsilon)^{k-1},$$
$$\sum_\alpha |[(\delta V)^k]_{b\,\beta}^{a\,\alpha}| \le (\epsilon)^k, \tag{A.22}$$

where $\eta$ and $\epsilon$ are small. We will prove in Sections (A.7) and (A.6) that

$$\epsilon_{\mathrm{MM}} = 2\frac{3}{\sqrt{2}}\frac{2S_0}{N^{1/10}}, \quad \eta_{\mathrm{MM}} = 24\frac{3}{\sqrt{2}}\frac{2^3}{N^{3/10}}, \tag{A.23}$$

$$\epsilon_{\mathrm{CTKT}} = \frac{22 S_0^5}{N}, \quad \eta_{\mathrm{CTKT}} = \frac{512}{N}. \tag{A.24}$$

## A.4  Gram matrix bounds

This Section is dedicated to the following technical Corollary:

**Corollary 1.** *Assume $\epsilon < 1$. Then we have the following bounds:*

- Sum of the diagonal elements:

$$|\sum_\alpha \delta M_{s\,\alpha}^{s\,\alpha}| \le \sum_\alpha \sum_{k=2}^{+\infty} |C_{1/2}^k (\delta V^k)_{s\,\alpha}^{s\,\alpha}| \tag{A.25}$$

$$\le 2^{S_0}\eta \sum_{k=2}^{+\infty} |C_{1/2}^k|\epsilon^{k-1} = 2^{S_0}\eta\frac{1}{\epsilon}\left(1 - \sqrt{1-\epsilon} - \frac{\epsilon}{2}\right) \le \frac{\epsilon}{2}2^{S_0}\eta. \tag{A.26}$$

Factor $2^{S_0}$ comes from summing over $\alpha$.

- Similar sum of diagonal elements squared:

$$|\sum_\alpha \delta M_{s\,\alpha}^{s\,\alpha}\delta M_{s\,\alpha}^{s\,\alpha}| \le 2^{S_0}(\eta)^2 \sum_{k,l=2} |C_{1/2}^k C_{1/2}^l|\epsilon^{k+l-2} \tag{A.27}$$

$$= 2^{S_0}(\eta)^2\left(\frac{1-\sqrt{1-\epsilon}-\epsilon/2}{\epsilon}\right)^2 \le \frac{\epsilon^2}{4}2^{S_0}(\eta)^2. \tag{A.28}$$

- Even if we have a sum over singlets states $|a\rangle\langle a|$ we can use the definition of $\delta M$ and turn it into identity:

$$\sum_{a,\alpha} \delta M^{s\ \alpha}_{a\ \alpha} \delta M^{a\ \alpha}_{s\ \alpha} =$$

$$\sum_{a,\alpha} \sum_{\gamma,\mu;k,l=2} C^k_{1/2} C^l_{1/2} \langle s|\mathcal{O}_{\alpha\gamma_1}\dots\mathcal{O}_{\gamma_{k-1}\alpha}|a\rangle\langle a|\mathcal{O}_{\alpha\mu_1}\dots\mathcal{O}_{\mu_{l-1}\alpha}|s\rangle. \quad \text{(A.29)}$$

We can insert more projectors onto code subspace to turn this sum into matrix powers of $\delta V$, since

$$(\delta V)^{a\ \alpha}_{b\beta} = \langle a|U^\dagger_\alpha U_\beta|b\rangle = \langle a|\mathcal{O}_{\alpha\beta}|b\rangle. \quad \text{(A.30)}$$

For two norms involving $\mathcal{O}_{\alpha\gamma_1}$ and $\mathcal{O}_{\alpha\mu_1}$ we will use the basic bound with $\eta$ (Corollary 1, $k = 1$), so that we can freely sum over all $\gamma$ and $\mu$:

$$|\sum_{a,\alpha} \delta M^{s\ \alpha}_{a\ \alpha} \delta M^{a\ \alpha}_{s\ \alpha}| \leq (\eta)^2\, 2^{S_0} \sum_{k,l=2} |C^k_{1/2} C^l_{1/2}|\epsilon^{k+l-2}$$

$$= (\eta)^2\, 2^{S_0} \left(\frac{1-\sqrt{1-\epsilon}-\epsilon/2}{\epsilon}\right)^2 \leq \frac{\epsilon^2}{4}(\eta)^2\, 2^{S_0}. \quad \text{(A.31)}$$

- We will need to bound the sum involving two set of fermionic strings, $\alpha$ and $\beta$:

$$\sum_{\alpha,\beta} \delta M^{s\ \alpha}_{s\ \beta} \delta M^{s\ \beta}_{s\ \alpha} = \sum_{\alpha,\beta;k,l=1} C^k_{1/2} C^l_{1/2} (\delta V^k)^{s\ \alpha}_{s\ \beta}(\delta V^l)^{s\ \beta}_{s\ \alpha}. \quad \text{(A.32)}$$

Again, expanding each $\delta V$ in terms of norms $\mathcal{O}_{\gamma_i\gamma_{i+1}}$, we use the $\eta$ bound for the norm with $\mathcal{O}_{\alpha\gamma_1}$. After that we can perform the sum over all index strings including $\beta$. Notice that now the sum over $k, l$ starts from 1:

$$|\sum_{\alpha,\beta} \delta M^{s\ \alpha}_{s\ \beta} \delta M^{s\ \beta}_{s\ \alpha}| \leq \eta 2^{S_0} \sum_{k,l=1} |C^k_{1/2} C^l_{1/2}|\epsilon^{k+l-1} \quad \text{(A.33)}$$

$$= \eta 2^{S_0} \frac{1}{\epsilon}\left(1-\sqrt{1-\epsilon}\right)^2 \leq \epsilon\eta 2^{S_0}.$$

- Using the same reasoning we straightforwardly obtain the following bounds for various products of three $\delta M$:

$$|\sum_{\alpha,\beta,a} \delta M^{s\ \alpha}_{a\ \alpha} \delta M^{a\ \alpha}_{s\ \beta} \delta M^{s\ \beta}_{s\ \alpha}| \leq 2^{S_0}(\eta)^2 \frac{1}{\epsilon^2}\left(1-\sqrt{1-\epsilon}-\epsilon/2\right)\left(1-\sqrt{1-\epsilon}\right)^2$$

$$\leq \frac{\epsilon^2}{2} 2^{S_0}(\eta)^2, \quad \text{(A.34)}$$

$$|\sum_{\alpha,a} \delta M^{s\ \alpha}_{a\ \alpha} \delta M^{a\ \alpha}_{s\ \alpha} \delta M^{s\ \alpha}_{s\ \alpha}| \leq 2^{S_0}(\eta)^3 \left(\frac{1-\sqrt{1-\epsilon}-\epsilon/2}{\epsilon}\right)^3 \leq \frac{\epsilon^3}{8} 2^{S_0}(\eta)^3. \quad \text{(A.35)}$$

- And finally, the sum involving four $\delta M$:

$$|\sum_{a,b,\alpha,\beta} \delta M^{s\ \alpha}_{a\ \alpha} \delta M^{a\ \alpha}_{s\ \beta} \delta M^{s\ \beta}_{b\ \alpha} \delta M^{b\ \alpha}_{s\ \alpha}| \leq 2^{S_0}(\eta)^3\, \epsilon\left(\frac{1-\sqrt{1-\epsilon}}{\epsilon}\right)^2\left(\frac{1-\sqrt{1-\epsilon}-\epsilon/2}{\epsilon}\right)^2$$

$$\leq \frac{\epsilon^3}{4} 2^{S_0}(\eta)^3. \quad \text{(A.36)}$$

## A.5 Finishing the proof

Having obtained the orthogonal basis $|\widetilde{a\ \alpha}\rangle$, we can defined enhanced projectors:

$$\widetilde{P}_\alpha = \sum_a |\widetilde{a\ \alpha}\rangle\langle\widetilde{a\ \alpha}|. \tag{A.37}$$

And the projector orthogonal to all of them:

$$\widetilde{P}_R = \mathbf{1} - \sum_\alpha \widetilde{P}_\alpha. \tag{A.38}$$

We have to introduce it to make the following recovery operation well-defined, but it is not going to play any significant role, since it is orthogonal to all $|a\rangle$ and $|a\ \alpha\rangle$.

We define the enhanced recovery operation $\widetilde{\mathcal{R}}$ as:

$$\widetilde{\mathcal{R}}(\sigma) = \sum_\alpha U^\dagger \widetilde{P}_\alpha \sigma \widetilde{P}_\alpha U_\alpha + \widetilde{P}_R \sigma \widetilde{P}_R. \tag{A.39}$$

**Theorem 7.** *Suppose that the Gram matrix $\delta V$ obeys the bounds (A.22) with $\epsilon, \eta < 1$. Then the recovery operation $\widetilde{\mathcal{R}}$ has following lower bound on fidelity*

$$F(\rho, \widetilde{\mathcal{R}}(\mathcal{E}(\rho))) \geq 1 - 6\epsilon\eta \tag{A.40}$$

*for $\rho$ in singlet subspace.*

*Proof.* Assume $\rho = |s\rangle\langle s|$. Then the fidelity is

$$F^2_{|s\rangle\langle s|} = F\left(|s\rangle\langle s|, \widetilde{\mathcal{R}}\left(\mathcal{E}\left(|s\rangle\langle s|\right)\right)\right)^2 = \frac{1}{M_0} \sum_{\alpha\beta} \langle s|U^\dagger_\alpha \widetilde{P}_\alpha U_\beta|s\rangle\langle s|U^\dagger_\beta \widetilde{P}_\alpha U_\alpha|s\rangle$$

$$= \frac{1}{2^{S_0}} \sum_{\alpha,\beta;a,b} M^{s\ \alpha}_{a\ \alpha} M^{a\ \alpha}_{s\ \beta} M^{s\ \beta}_{b\ \alpha} M^{b\ \alpha}_{s\ \alpha}. \tag{A.41}$$

Now we expand $M = \mathbf{1} + \delta M$. The term with four $\mathbf{1}$ yield 1, the rest of the terms we can bound with Corollary 1. Putting all the terms together we have:

$$|1 - F^2_{|s\rangle\langle s|}| \leq 3\epsilon\,(\eta) + \left(5 \times \frac{\epsilon^2}{4} + 2 \times \frac{\epsilon^2}{2}\right)(\eta)^2 + \left(2 \times \frac{\epsilon^3}{8} + \frac{\epsilon^3}{4}\right)(\eta)^3 \leq 6\epsilon\eta. \tag{A.42}$$

The last inequality holds for $\epsilon, \eta < 1$.

The last step is to consider a generic density matrix

$$\rho = \sum_i p_i \rho_i = \sum_i p_i |s_i\rangle\langle s_i|. \tag{A.43}$$

Then after the recovery the density matrix $\widetilde{\mathcal{R}}(\mathcal{E}(\rho))$ is given by:

$$\widetilde{\mathcal{R}}(\mathcal{E}(\rho)) = \sum_i p_i \widetilde{\rho}_i = \sum_i p_i \widetilde{\mathcal{R}}(\mathcal{E}(\rho_i)). \tag{A.44}$$

To obtain the desired bound we use the strong concavity of the fidelity:

$$F\left(\rho, \widetilde{\mathcal{R}}(\mathcal{E}(\rho))\right) = F\left(\sum_i p_i, \rho_i, \sum_i p_i \widetilde{\rho}_i\right) \geq \sum_i p_i F(\rho_i, \widetilde{\rho}_i) \geq 1 - 6\epsilon\eta. \tag{A.45}$$

$\square$

## A.6 Specific bounds for CTKT model

### A.6.1 Basic bounds

Let us begin from recalling the basic bounds obtained in the main text.

- For CTKT model, Majorana fermion index $I = (a, b, c)$, where $a, b, c$ are fundamental indices of $O(N)^3$.

- If the total length of $E_\alpha^\dagger E_\beta$ is $2k$, then the matrix elements of the singlet operator $\mathcal{O}_{\alpha\beta}$ are bounded by:

$$|\langle s|\mathcal{O}_{\alpha\beta}|s\rangle| \le \frac{8^k k!^3}{N^{k/2}}, \tag{A.46}$$

where state $|s\rangle$ is $O(N)^3$ singlet.

- If function $l(\alpha)$ denote the length of a string $\alpha$, the above bound can be rewritten as:

$$|\langle s|\mathcal{O}_{\alpha\beta}|s\rangle| \le \frac{8^{l/2}(l/2)!^3}{N^{l/4}} \le \frac{512}{N}, \; l = l(\alpha\beta). \tag{A.47}$$

The right hand side is a decreasing function of $l$ since we are considering $l \le S_0 \le N^{1/6}/2$. Coefficient 512 comes from $l = 4$.

### A.6.2 Gram matrix bounds

**Lemma 1.** *The following inequality holds for fixed $a, b, \alpha$:*

$$\sum_{\beta, \beta \ne \alpha} |(\delta V)_{b\ \beta}^{a\ \alpha}| = \sum_{\beta, \beta \ne \alpha} |\langle a\ \alpha|b\ \beta\rangle| \le \frac{22 S_0^5}{N} \tag{A.48}$$

*as long as $S_0 \le N^{1/6}/2$. Also in the above expression we used the bound for singlet operators in the form (A.47).*

*Proof.* It is not difficult to see that above sum goes over all non-empty strings $\gamma$, regardless of the string $\alpha$:

$$\sum_{\beta, \beta \ne \alpha} |\langle a\ \alpha|b\ \beta\rangle| = \sum_{\gamma, \gamma \ne \emptyset} |\langle a|U_\gamma|b\rangle| \le \sum_{k=2}^{S_0/2} C_{S_0}^{2k} \frac{8^k k!^3}{N^{k/2}}. \tag{A.49}$$

Notice that although $\gamma$ is a combination of strings $\alpha$ and $\beta$ it may not be longer than $S_0$. The last inequality we used comes from eq. (A.46). Binomial coefficient comes from choosing $2k$ fermionic sites from available $S_0$. It is easy to see that as long as $S_0 \le N^{1/6}/2$ the terms in the sum are decreasing with $k$. So we can bound the sum by taking the biggest term $k = 2$ and multiplying it by $S_0/2$. This way we obtain the bound (A.48). This seems like a very crude approximation, but in fact as can be seen from Stirling approximation, for $k \propto e N^{1/6}$, a single matrix element $8^k k!^3/N^{k/2}$ becomes of order one, so $S_0 \lesssim \mathcal{O}(N^{1/6})$ is a tight bound in terms of powers of $N$.

$\square$

Recall that the bound (A.46) or equivalently (A.47) is true for any singlet state $|c\rangle$. Also recall that $\mathcal{O}_{\alpha\beta}$ is either hermitian or anti-hermitian. It implies that it is actually a bound for the (absolute value of) maximal eigenvalue. From that we infer that

$$\sqrt{\langle s|\mathcal{O}_{\alpha\beta}^\dagger \mathcal{O}_{\alpha\beta}|s\rangle} \le \frac{(l/2)!^3 8^{l/2}}{N^{l/4}}, \; l = l(\alpha\beta). \tag{A.50}$$

Again we would like to stress that this bound hold *for any* singlet $|s\rangle$.

Now we will use this fact to bound $\delta V^k$ for arbitrary $k$. Let us recall that $\delta V$ is the Gram matrix without diagonal elements:

$$\delta V^{a\ \alpha}_{b\ \beta} = \langle a\ \alpha|b\ \beta\rangle - \delta^{a\ \alpha}_{b\ \beta}. \tag{A.51}$$

**Lemma 2.** *For CTKT model*

$$|[(\delta V)^k]^{a\ \alpha}_{b\ \beta}| \le \frac{512}{N}\left(\frac{22S_0^5}{N}\right)^{k-1},$$

$$\sum_\alpha |[(\delta V)^k]^{a\ \alpha}_{b\ \beta}| \le \left(\frac{22S_0^5}{N}\right)^k. \tag{A.52}$$

*Proof.* Explicitly the above matrix element is:

$$[(\delta V)^k]^{a\ \alpha}_{b\ \beta} = \sum_{(c_1\gamma_1)\dots,(c_{k-1}\gamma_{k-1})} \langle a|U_\alpha^\dagger U_{\gamma_1}|c_1\rangle\langle c_1|\dots|c_{k-1}\rangle\langle c_{k-1}|U_{\gamma_{k-1}}^\dagger U_\beta|b\rangle. \tag{A.53}$$

Note that since singlet states $|a\rangle$ are orthogonal, the sum over the fermionic strings goes over $\alpha \ne \gamma_1 \ne \gamma_2 \ne \dots \ne \gamma_{k-1} \ne \beta$.

In each matrix element $\langle c_i|U_{\gamma_i}^\dagger U_{\gamma_{i+1}}|c_{i+1}\rangle$ we can substitute $U_{\gamma_i}^\dagger U_{\gamma_{i+1}}$ by the singlet operator $\mathcal{O}_{\gamma_i,\gamma_{i+1}}$. After that the sum over $c_i$ converts $|c_i\rangle\langle c_i|$ into identity operator. Here it is important that the code subspace coincides with the whole singlet space. So we get

$$[(\delta V)^k]^{a\ \alpha}_{b\ \beta} = \sum_{\gamma_1 \ne \gamma_2 \ne \dots \ne \gamma_{k-1}} \langle a|\mathcal{O}_{\alpha,\gamma_1}\dots\mathcal{O}_{\gamma_{k-1},\beta}|b\rangle. \tag{A.54}$$

At first sight it seems that for very large $k$ we are dealing with a very long string of fermionic operators, which expectation value might not be suppressed by $1/N$. But in fact we have a product of well-known singlet operators $\mathcal{O}_{\alpha\beta}$ for which we have a nice bound given by Lemma 1. We can rewrite the matrix element in (A.54) as:

$$\sqrt{\langle a|\mathcal{O}_{\alpha,\gamma_1}^\dagger \mathcal{O}_{\alpha,\gamma_1}|a\rangle}\langle a\ \mathcal{O}_{\alpha,\gamma_1}|\mathcal{O}_{\gamma_2,\gamma_3}\dots\mathcal{O}_{\gamma_{k-1},\beta}|b\rangle, \tag{A.55}$$

where $|a\ \mathcal{O}_{\alpha,\gamma_1}\rangle$ is a normalized singlet state given by:

$$|a\ \mathcal{O}_{\alpha,\gamma_1}\rangle = \mathcal{O}_{\alpha,\gamma_1}|a\rangle\frac{1}{\sqrt{\langle a|\mathcal{O}_{\alpha,\gamma_1}^\dagger \mathcal{O}_{\alpha,\gamma_1}|a\rangle}}. \tag{A.56}$$

So we could lower the number of singlet operators by introducing the state $|a\ \mathcal{O}_{\alpha,\gamma_1}\rangle$.

Repeating this procedure for other operators and using the bound (A.50) we obtain:

$$|[(\delta V)^k]^{a\ \alpha}_{b\ \beta}| \le \sum_{\alpha \ne \gamma_1 \ne \dots \ne \gamma_{k-1} \ne \beta} f(\alpha\gamma_1)f(\gamma_1\gamma_2)\dots f(\gamma_{k-1}\beta), \tag{A.57}$$

$$f(\mu\nu) = \frac{8^{l(\mu\nu)/2}(l(\mu\nu)/2)!^3}{N^{l(\mu\nu)/4}}.$$

If we do not have a sum over $\alpha$, we can use the bound (A.46) for $\alpha\gamma_1$ pair and after repeatedly using Lemma 1 for $\gamma_1,\dots,\gamma_{k-1}$ we get the statement of the Lemma. Otherwise we start the sum from $\alpha$, yielding extra power of $(22S_0^5)/N$. $\qquad\square$

## A.7  Specific bounds for matrix models

### A.7.1  Basic bounds

- For matrix models, Majorana index $I$ is $I = (a, (ij))$ Where $(ij)$ is $SU(N)$ adjoint index and $a$ is arbitrary "flavor" index taking $D$ values.

- If the total length of $E_\alpha^\dagger E_\beta$ is $k$, then the matrix element of $\mathcal{O}_{\alpha\beta}$ in a (normalized) singlet state $|s\rangle$ is bounded by

$$|\langle s|\mathcal{O}_{\alpha\beta}|s\rangle| \leq \frac{3}{\sqrt{2}} \frac{k! 2^k}{N^{k/10}} \leq 24 \frac{3}{\sqrt{2}} \frac{2^3}{N^{3/10}}, \tag{A.58}$$

where the last inequality comes from putting $k = 3$ and demanding

$$k \leq N^{1/10}. \tag{A.59}$$

- If function $l(\alpha)$ denote the length of a string $\alpha$, then one can rewrite the above bound as:

$$|\langle s|\mathcal{O}_{\alpha\beta}|s\rangle| \leq \frac{3}{\sqrt{2}} \frac{l! 2^l}{N^{l/10}}, \quad s = s(\alpha\beta). \tag{A.60}$$

### A.7.2  Gram matrix bounds

**Lemma 3.** *The following inequality holds for fixed $a, b, \alpha$:*

$$\sum_{\beta, \beta \neq \alpha} |(\delta V)_{b\ \beta}^{a\ \alpha}| = \sum_{\beta, \beta \neq \alpha} |\langle a\ \alpha|b\ \beta\rangle| = \sum_{\beta, \beta \neq \alpha} |\langle a|\mathcal{O}_{\alpha\beta}|b\rangle|$$

$$\leq \frac{3}{\sqrt{2}} \sum_{l=3}^{S_0} C_{S_0}^l 2^l \frac{l!}{N^{l/10}} \leq 2 \frac{3}{\sqrt{2}} \frac{2 S_0}{N^{3/10}}, \quad s = s(\alpha\beta), \tag{A.61}$$

*as long as the lengths of $\alpha, \beta$ are less than $\leq N^{1/10}$. Also in the above expression we used the bound for singlet operators in the form (A.60).*

*Proof.* One can use the fact that the sum

$$\frac{1}{S_0!} \sum_{l=3}^{S_0} C_{S_0}^l \frac{l! 2^l}{N^{l/10}} = \sum_{l=3}^{S_0} \frac{1}{(S_0 - l)! M^l}, \quad M = \frac{N^{1/10}}{2}, \tag{A.62}$$

is equal to

$$\frac{e^M M^{-S_0} \Gamma(1 + S_0, M)}{S_0!} - \frac{1}{S_0!} - \frac{1}{M(S_0 - 1)!} - \frac{1}{M^2(S_0 - 2)!}, \tag{A.63}$$

where $\Gamma(1 + S_0, M)$ is incomplete Gamma-function:

$$\Gamma(1 + S_0, M) = \int_M^{+\infty} t^{S_0} e^{-t} dt, \tag{A.64}$$

for which the following bounds exist [57]:

$$M^{S_0} e^{-M} \leq |\Gamma(1 + S_0, M)| \leq C M^{S_0} e^{-M}, \tag{A.65}$$

where $C$ is a number such that $M > \frac{C}{C-1} S_0$. In our case we can put $C = 1 + 2S_0/M$ if $M > 2S_0$. This way we obtain the bound

$$\sum_{\beta, \beta \neq \alpha} |(\delta V)_{b\ \beta}^{a\ \alpha}| \leq 2 \frac{3}{\sqrt{2}} \frac{2 S_0}{N^{3/10}}, \tag{A.66}$$

as long as $S_0 \leq N^{1/10}$. $\qquad\square$

Also recall that $\mathcal{O}_{\alpha\beta}$ is either hermitian or anti-hermitian. It implies that it is actually a bound for the (absolute value of) maximal eigenvalue. From that we infer that

$$\sqrt{\langle c|\mathcal{O}_{\alpha\beta}^{\dagger}\mathcal{O}_{\alpha\beta}|c\rangle} \leq \frac{3}{\sqrt{2}} 2^{s(\alpha\beta)} \frac{s(\alpha\beta)!}{N^{s(\alpha\beta)/10}}. \tag{A.67}$$

Now we will use this fact to bound $\delta V^k$ for arbitrary $k$. Let us recall that $\delta V$ is the Gram matrix without diagonal elements:

$$\delta V_{b\,\beta}^{a\,\alpha} = \langle a\ \alpha|b\ \beta\rangle - \delta_{b\,\beta}^{a\,\alpha}. \tag{A.68}$$

**Lemma 4.** *For fermionic matrix models*

$$|[(\delta V)^k]_{b\,\beta}^{a\,\alpha}| \leq 24\frac{3}{\sqrt{2}}\frac{2^3}{N^{3/10}}\left(2\frac{3}{\sqrt{2}}\frac{2S_0}{N^{1/10}}\right)^{k-1}, \tag{A.69}$$

$$\sum_{\alpha}|[(\delta V)^k]_{b\,\beta}^{a\,\alpha}| \leq \left(2\frac{3}{\sqrt{2}}\frac{2S_0}{N^{1/10}}\right)^{k}. \tag{A.70}$$

*Proof.* Again, the proof is almost the same as in the CTKT case (Lemma 2). We again have to use the property that we can transform the matrix element

$$\langle a|\mathcal{O}_{\alpha\beta}\mathcal{O}_{\gamma\delta}|b\rangle \tag{A.71}$$

into

$$\langle\widetilde{a}|\mathcal{O}_{\gamma\delta}|b\rangle, \tag{A.72}$$

where singlet state $|\widetilde{a}\rangle \propto \mathcal{O}_{\alpha\beta}^{\dagger}|a\rangle$. $\qquad\square$

# B   Bounding operators in matrix models

## B.1   Main argument

In this Section we will prove the following bounds on singlet fermionic operators $\mathcal{O}_k$ made from $3 \leq k \leq 2\sqrt{N}$ fermions:

$$\begin{aligned}|\mathcal{O}_{3m}| &\leq 2^m N^{5m/2},\\ |\mathcal{O}_{3m+1}| &\leq \sqrt{2}N \times 2^m N^{5m/2},\\ |\mathcal{O}_{3m+2}| &\leq 2N^2 \times 2^m N^{5m/2},\\ m &\geq 1,\end{aligned} \tag{B.1}$$

and

$$|\mathcal{O}_4| \leq 2N^3. \tag{B.2}$$

In this Section, "bounds" mean bounds on matrix elements in singlet states. In other words,

$$|\mathcal{O}| \leq C \Longleftrightarrow |\langle s_1|\mathcal{O}|s_2\rangle| \leq C, \tag{B.3}$$

for any normalized singlet states $|s_{1,2}\rangle$.

It would be useful to introduce auxiliary operators $Q_{ij}^{a_1 a_2}$ as:

$$Q_{ij}^{a_1 a_2} = \sum_k \psi_{ik}^{a_1}\psi_{kj}^{a_2}. \tag{B.4}$$

One simple, but central fact is that the operator $\text{Tr}(Q^{aa}Q^{aa}) = \text{Tr}(\psi^a)^4$ is proportional to identity operator:

$$\text{Tr}(\psi^a)^4 = (N^2 - 1)\left(2N - \frac{1}{N}\right) \times \mathbf{1} \leq 2N^3 \times \mathbf{1}. \tag{B.5}$$

One can see this by commuting the first $\psi$ to the last position and using the cyclicity of the trace. This agrees with group-theoretic expectations: operators $Q^{aa}$ are proportional to generators of $SU(N)_a$. This group rotates fermions $\psi^a$ only. Because they anti-commute the corresponding representations include Young diagrams with rows and columns of length no more than $N$. It means that the quadratic Casimir is bounded by const $\times N^3$. The quadratic Casimir is proportional to $\text{Tr}(Q^{aa}Q^{aa})$.

We can obtain a similar bound for other four-fermion operators. Suppose we have $\text{Tr}(\psi^1\psi^2\psi^3\psi^4)$. This operator is not hermitian, so we will bound the hermitian part. The anti-hermitian part can be bounded using exactly the same approach. The hermitian part is $\text{Tr}(\psi^1\psi^2\psi^3\psi^4)/2 + \text{Tr}(\psi^4\psi^3\psi^2\psi^1)/2$. For any two set of operators $A_I, B_I$, there is the following inequality:

$$0 \leq \sum_I (A_I + B_I)(A_I + B_I)^\dagger = \sum_I A_I A_I^\dagger + B_I B_I^\dagger + A_I B_I^\dagger + B_I A_I^\dagger. \tag{B.6}$$

In our case $A_I \equiv A_{ij} = (\psi^1\psi^2)_{ij}$ and $B_{ij}^\dagger = (\psi^3\psi^4)_{ij}$. Hence the hermitian part can be bounded by

$$\frac{1}{2}\text{Tr}(\psi^1\psi^2\psi^2\psi^1) + \frac{1}{2}(\psi^4\psi^3\psi^3\psi^4). \tag{B.7}$$

Repeating this procedure one more time we see that any four-fermion operator can be bounded by $2N^3$.

Let us first consider a single-trace operator $\mathcal{O}_{3m}$ containing $3m$ fermions. We can write it as product of the form $Q\psi Q\psi Q\psi \ldots$:

$$\mathcal{O}_{3m} = \sum_{ijk\ldots} Q_{ij}^{a_1a_2}\psi_{jk}^{a_3}Q_{kl}^{a_4a_5}\ldots\psi_{ni}^{a_{3m}}. \tag{B.8}$$

We are interested in evaluating the matrix element between the singlet states $|s_{1,2}\rangle$:

$$\langle s_1|\mathcal{O}_{3m}|s_2\rangle. \tag{B.9}$$

Our main tool is the following trick first used in [30]. Since $|s_{1,2}\rangle$ is invariant under any $SU(N)$ rotations we can force some of the indices to have specific values. For example, for each separate term with some $i$, we can use $SU(N)$ rotation to make it $i = 1$. So we can put $i = 1$ everywhere and multiply the sum by $N$. We can continue doing this for other indices. However, for index $j$ now we have two choices: either $j = 1$ and we leave it alone or $j \neq 1$ and using a different $SU(N)$ rotation we can make it $j = 2$. Notice that since $j \neq 1$ this additional rotation does not change the value of $i = 1$. In the second case we multiply the sum by $N - 1$. We can continue doing this. As long as $2m < N$ the sum over indices is dominated by configurations where all indices are different. We will give an additional argument for this below eq. (B.14). So $SU(N)$ rotations can transform it into

$$\langle s_1|\mathcal{O}_{3m}|s_2\rangle = N^{2m}\langle s_1|Q_{12}^{a_1a_2}\psi_{23}^{a_3}Q_{34}^{a_4a_5}\ldots\psi_{2m,1}^{a_{3m}}|s_2\rangle, \tag{B.10}$$

up to subleading $1/N$ corrections. Now we can use Cauchy–Schwartz–Popov inequality.[22] We will have a product of $\psi_{2m,1}^{a_{3m}}\psi_{1,2m}^{a_{3m}}$ in the middle. For the reasons explained in Section 4.1 this

---

[22] For arbitrary states $|\xi_{1,2}\rangle$ and operator $A$:

$$|\langle\xi_1|A|\xi_2\rangle|^2 = \langle\xi_1|A|\xi_2\rangle\langle\xi_2|A^\dagger|\xi_1\rangle \leq \sum_{|\xi_2\rangle}\langle\xi_1|A|\xi_2\rangle\langle\xi_2|A^\dagger|\xi_1\rangle = \langle\xi_1|AA^\dagger|\xi_1\rangle.$$

This is standard Cauchy–Schwartz inequality. Popov part was specifying indices in eq. (B.10).

product is not proportional to identity operator. However it is a positive hermitian operator and its eigenvalues can be bounded[23] by 2. Moreover, we have two hermitian conjugate operators on the sides, so

$$\langle s|H\psi_{2m,1}^{a_{3m}}\psi_{1,2m}^{a_{3m}}H^\dagger|s\rangle \le 2\langle s|HH^\dagger|s\rangle. \tag{B.11}$$

Moreover, due to results of Section B.2, we can commute conjugated $\psi$ to the middle ignoring arising anticommutators. Repeating this many times, in the end we obtain:

$$|\langle s|\mathcal{O}_{3m}|s\rangle|^2 \le N^{4m}2^m\langle s|Q_{12}^{a_1 a_2}Q_{21}^{a_2 a_1}\ldots|s\rangle. \tag{B.12}$$

We can convert the right hand side into a singlet operator:

$$|\langle s|\mathcal{O}_{3m}|s\rangle|^2 \le N^{2m}2^m\langle s|\operatorname{Tr}(Q^{a_1 a_2}Q^{a_2 a_1})\operatorname{Tr}(Q^{a_3 a_4}Q^{a_4 a_3})\ldots|s\rangle. \tag{B.13}$$

Operator $\operatorname{Tr}(Q^{a_1 a_2}Q^{a_2 a_1})$ is four-fermion, so it can be bounded by $2N^3$. Therefore in the end we get

$$|\langle s|\mathcal{O}_{3m}|s\rangle|^2 \le N^{2m}\left(4N^3\right)^m. \tag{B.14}$$

Now let us explain why cases with coincident indices in eq. (B.8) are subleading in $N$. Essentially every coincidence will bring to eq. (B.10) an extra term with $1/N$ suppression and slightly different singlet operator spectrum. For example, leading to operators like $\operatorname{Tr}(Q^{a_1 a_2}Q^{a_3 a_4}Q^{a_5 a_6}Q^{a_7 a_8})$ instead of $\operatorname{Tr}(Q^{a_1 a_2}Q^{a_3 a_4})\operatorname{Tr}(Q^{a_5 a_6}Q^{a_7 a_8})$ in eq. (B.13). For $l$ coincidences, the number of such extra operators can be bounded by $l!2^{l-1}$ since every coincidence can be used to produce a different singlet and also we can contract coincident indices in different ways.

The scaling of these operators will be the same as in the case of no coincidences. Let us explain this in more detail. Unless we have more than $3m/2$ coincidences, the operators appearing there will have less than $3m$ operators, so we can use induction and (B.1) to bound them. If we have $3m/2$ coincidences or more, then these contributions can be bounded by $N^{3m/2}$(the number of terms in the sum), which is already better then (B.1).

Also, coincidences can occur in many different places, leading to an extra binomial coefficient $C_{2m}^l$. In the end, the relative contribution of coincident configurations is

$$\sum_{l=1}^{2m}C_{2m}^l\frac{2^{l-1}l!}{N^l} \le \sum_{l=1}^{\infty}\frac{(2m)^l 2^{l-1}}{N^l} = \frac{1}{2}\left(\frac{1}{1-\frac{4m}{N}}-1\right), \tag{B.15}$$

which is small since we are interested in $3m \le N^{1/4}$.

Now let us consider a single-trace operator with uneven number of fermions. One can split one fermion:

$$\mathcal{O}_{3m+1} = \sum_{ij}\mathcal{O}_{3m}^{ij}\psi_{ji}^a. \tag{B.16}$$

If we are studying an expectation value in some singlet state $|s\rangle$ we can use $SU(N)$ rotation to make index $j$ to be 1. Then there are two possibilities: either $i$ is 1 or not 1. If it is not, we can use another $SU(N)$ rotation to make it equal 2:

$$\mathcal{O}_{3m+1} = \sum_{ij}\mathcal{O}_{3m}^{ij}\psi_{ji}^a = N(N-1)\mathcal{O}_{3m}^{12}\psi_{12}^a + N\mathcal{O}_{3m}^{11}\psi_{11}^a. \tag{B.17}$$

---

[23]One can see this by writing

$$\psi_{12}^a\psi_{21}^a = \sum_{(ij),(kl)}T_{12}^{(ij)}T_{21}^{(kl)}\psi_{(ij)}^a\psi_{(kl)}^a.$$

$SU(N)$ generators can be chosen so that matrix $T^{(ij)}$ has $1/\sqrt{2}$ or $\pm i/\sqrt{2}$ in $ij$ and $ji$ positions and the rest of the elements are zero. So the sum over $(ij),(kl)$ contains only 4 elements. Operators $\psi_{(ij)}^a$ square to one, so the whole sum can be bounded by 2. If we have a coincidence of indices, then we do not have to introduce the generators, since $\psi_{ii}^a$ actually square to identity.

We can again use Cauchy–Schwartz inequality. In our case we have two pairs: $\mathcal{O}_{3m}^{12}$, $\psi_{12}^{a}$ and $\mathcal{O}_{3m}^{11}$, $\psi_{11}^{a}$. For the first pair we get

$$|\langle \mathcal{O}_{3m}^{12} \psi_{12}\rangle|^2 \leq 2\langle \mathcal{O}_{3m}^{12} \mathcal{O}_{3m}^{\dagger,21}\rangle = \frac{2}{N^2}\langle \mathrm{Tr}\, \mathcal{O}_{3m} \mathcal{O}_{3m}^{\dagger}\rangle, \tag{B.18}$$

which is single-trace operator with $6m$ fermions, which we can bound with $4^m N^{5m}$. For the second pair we get a similar expression:

$$|\langle \mathcal{O}_{3m}^{11} \psi_{11}\rangle|^2 \leq 2\langle \mathcal{O}_{3m}^{11} \mathcal{O}_{3m}^{\dagger,11}\rangle = \frac{2}{N^2}\langle \mathrm{Tr}\, \mathcal{O}_{3m} \mathcal{O}_{3m}^{\dagger}\rangle, \tag{B.19}$$

we omitted $\mathrm{Tr}(\mathcal{O}_{3m})$ because in the original expression $\mathcal{O}_{3m}$ was convoluted with trace-free elementary fermion $\psi_{ij}^{a}$, so we could subtract the trace part from $\mathcal{O}_{3m}$. Combining the two contributions we get

$$\mathcal{O}_{3m+1} \leq N\sqrt{2\,\mathrm{Tr}\, \mathcal{O}_{3k} \mathcal{O}_{3m}^{\dagger}}. \tag{B.20}$$

Let us consider $\mathcal{O}_{3m+2}$. We can split two $\psi$ operators:

$$\mathcal{O}_{3m+2} = \sum_{ijk} \mathcal{O}_{3m}^{ij} \psi_{jk}^{a} \psi_{ki}^{b}. \tag{B.21}$$

Using the same logic as above, specifying $ijk$ to be some fixed indices and using Cauchy–Schwartz inequality, we conclude that we can bound the original $\mathcal{O}_{3m+2}$ by square root of some $\mathcal{O}_{6m}$ times $2N^2$:

$$\langle s|\mathcal{O}_{3m+2}|s\rangle \leq 2N^2\sqrt{\langle s|\,\mathrm{Tr}\, \mathcal{O}_{6m}|s\rangle}, \tag{B.22}$$

this proves the last inequality in (B.1).

## B.2 Resolving anti-commutators

Let us consider a single trace operator $\mathcal{O}_k$ made with $k$ fermions $\psi_{ij}^{a}$. Suppose we want to rearrange positions of operators $\psi$. In this subsection we are going to demonstrate that additional operators arising from anti-commutators are suppressed by $1/N$. Intuitively, it happens if $k$ is not too large, so that the index sum is dominated by different indices. Namely, we will show that $k \lesssim \sqrt{N}$. This implies that in rearranging $\psi$ operators inside the trace we can neglect all (anti-)commutators. Our task is greatly simplified by the fact that anti-commutators are $c-$numbers, so we will simply reduce the fermionic number and then use the proposed bound (B.1). However, we should keep in mind that there could be $k(k-1)/2$ places where $\psi$ failed to anti-commute, we have to take it into account in order to prove that anti-commutators are subleading. Also we have to watch carefully if our manipulations are going to introduce factors

$$\sum_{ij} \psi_{ij}^{a} \psi_{ji}^{a} = N^2 - 1, \tag{B.23}$$

as they potentially can spoil the scaling (B.1).

Essentially, there are two cases when anti-commutators appear, in other words indices "collide":

- Collision of $(i_1, j_1)$ and $(i_2, j_2)$
  In this case the extra anti-commutator term is

$$\{\psi_{i_1 j_1}^{a_1}, \psi_{i_2 j_2}^{a_1}\} = 2\left(\delta_{i_1 j_2}\delta_{j_1 i_2} - \delta_{i_1 j_1}\delta_{i_2 j_2}\frac{1}{N}\right). \tag{B.24}$$

So we simply have $\mathcal{O}_k \to k(k-1)\big(\mathcal{O}_{k-l-2}\mathcal{O}_l + \frac{1}{N}\widetilde{\mathcal{O}}_{k-2}\big)$. Where each $\mathcal{O}_\#$ is a single trace operator. $l$ is determined by the distance between $\psi$ operators under the trace. Could it

be that we have introduced a disconnected piece like (B.23)? It is not possible for $\widetilde{\mathcal{O}}_{k-2}$ since we have already bounded all $k = 4$ operators. It is possible for $\mathcal{O}_{k-l-2}\mathcal{O}_k$. But in this case we do not have to multiply by $k(k-1)$ since such occurrence might happen only twice for each $\psi$. So we have to multiply by $2k$ instead. So in the end we may have $\propto k^2\mathcal{O}_{k-2}$ or $2kN^2\mathcal{O}_{k-4}$. But it is still consistent with the scaling (B.1) as long as $k \lesssim N$.

- Collision of $(i_1, j)$ and $(j, i_2)$:

$$\sum_j \{\psi_{i_1 j}^{a_1}, \psi_{ji_2}^{a_1}\} = 2(N - 1/N)\delta_{i_1 i_2}. \tag{B.25}$$

In this case we have $\mathcal{O}_k \to 2k(N - 1/N)\mathcal{O}_{k-2}$. Notice that we have multiplied by $2k$ because the $\psi$ operators share the same index, so there are only $k$ pairs like that. Could it be that we have created a piece like (B.23)? Notice that now both $\psi$ belong to the same single trace operator. And $\mathcal{O}_{k-2}$ is single trace too. So $\mathcal{O}_{k-2}$ is disconnected only if $k = 4$. But we have already covered all four-fermion operators in the previous Section.

This last case will yield the stringiest constraint. Namely, if originally we had $\mathcal{O}_{3k+3}$, which scales as $2^{k+1}N^{5(k+1)/2}$ we will have $2kN\mathcal{O}_{3k+1}$ which scales as $2kN^2 2^k N^{5k/2}$. It is smaller if

$$k \leq 2N^{1/2}. \tag{B.26}$$

## B.3 Color factors

We want to bound expressions like

$$\mathcal{C}_k = \mathrm{Tr}\left(T^{(ij)_1} T^{(ij)_2} \ldots T^{(ij)_k}\right) \tag{B.27}$$

by

$$|\mathcal{C}_k| \leq \sqrt{2}. \tag{B.28}$$

We can do it again using elementary algebra and the completeness relation:

$$\sum_{(pq)} T_{ij}^{(pq)} T_{kl}^{(pq)} = \delta_{il}\delta_{jk} - \frac{1}{N}\delta_{ij}\delta_{kl}. \tag{B.29}$$

We will use induction in $k$: $\mathcal{C}_1 = 0$ and $\mathcal{C}_2 \leq 1$ due to normalization (4.3).
We can consider the square of $\mathcal{C}_k$ and take a sum over $(ij)_k$:

$$
\begin{aligned}
|\mathcal{O}_k|^2 &= \mathrm{Tr}\left(T^{(ij)_1} \ldots T^{(ij)_k}\right)\mathrm{Tr}\left(T^{(ij)_k} \ldots T^{(ij)_1}\right) \\
&\leq \sum_{(ij)_k} \mathrm{Tr}\left(T^{(ij)_1} T^{(ij)_2} \ldots T^{(ij)_k}\right)\mathrm{Tr}\left(T^{(ij)_k} \ldots T^{(ij)_1}\right) \\
&= \mathrm{Tr}\left(T^{(ij)_1} \ldots T^{(ij)_{k-1}} T^{(ij)_{k-1}} \ldots T^{(ij)_1}\right) - \frac{1}{N}\mathrm{Tr}\left(T^{(ij)_1} \ldots T^{(ij)_{k-1}}\right)\mathrm{Tr}\left(T^{(ij)_{k-1}} \ldots T^{(ij)_1}\right).
\end{aligned} \tag{B.30}
$$

The first term can be rewritten as

$$\mathrm{Tr}\left(H T^{(ij)_{k-1}} T^{(ij)_{k-1}} H^\dagger\right). \tag{B.31}$$

Now, $T^{(ij)}$ are hermitian and due to normalization (4.3) has eigenvalues smaller then one. Therefore the first term can be bounded by

$$\mathrm{Tr}\left(H T^{(ij)_{k-1}} T^{(ij)_{k-1}} H^\dagger\right) \leq \mathrm{Tr}\left(H H^\dagger\right). \tag{B.32}$$

Repeating this for other operators we arrive at

$$|\mathcal{C}_k|^2 \leq 1 + \frac{1}{N}|\mathcal{C}_{k-1}|^2. \tag{B.33}$$

Hence, all $\mathcal{C}_k$ can be bounded by

$$|\mathcal{C}_k|^2 \leq 1 + \sum_{l=1}^{+\infty} \frac{1}{N^l} \leq 1 + \frac{1}{N-1} \leq 2, \tag{B.34}$$

for $N \geq 2$.

### B.4 Wrong contractions

In this Section we are going to estimate the relative contribution of "wrong contractions" (like $\text{Tr}\left(\psi^1\psi^3\psi^2\right)/N^2$ in eq. (4.11)). Due to the bound (B.1) these operators have the same scaling with $N$. So a problem may arise if the number of such extra operator overcome the $1/N$ suppressions.

 We can bound their number from above,[24] by noticing that "wrong contractions" are given by all possible permutations of fermionic operators. A trivial permutation gives unsuppressed operator, whereas other permutations are suppressed by $N^p$, where $p$ is the number of permuted fermions. If we fix $p$, there are $!p$ non-trivial[25] permutations. We can bound this number simply by $p!$. So the relative contribution of "wrong contractions" can be bounded by

$$\sum_{p=2}^{k} C_k^p \frac{p!}{N^p} = k! \sum_{p=2}^{k} \frac{1}{N^p(k-p)!} = \frac{e^N \Gamma(1+k,N)}{N^k} - 1 - \frac{k}{N}, \tag{B.35}$$

where $\Gamma(k+1,N)$ is incomplete Gamma function:[26]

$$\Gamma(k+1,N) = \int_{N}^{+\infty} t^k e^{-t} dt, \tag{B.36}$$

which has the following bounds [57]:

$$N^k e^{-N} \leq \Gamma(k+1,N) \leq C N^k e^{-N}, \tag{B.37}$$

where constant $C$ is such that $N \geq \frac{C}{C-1}k$. In our case assuming $k \leq N/2$ we choose it to be $C = 1 + 2k/N$. We conclude that the sum (B.35) can be bounded by

$$(\text{eq. (B.35)}) \leq \frac{k}{N} \leq \frac{1}{2}. \tag{B.38}$$

## C  Bounding singlet operators in CTKT model

In this Appendix we will prove the bound (5.9).

### C.1  Hamiltonian and Casimirs

As a warm-up, let us repeat the calculation from [30] and bound[27] the Hamiltonian by $N^5$. This is illustrated by Figure 2.

---

[24]It is an upper bound because we are neglecting the cyclicity of trace, thus adding extra operators

[25]Number $!p$, known as *derangement*, is the number of permutations of $p$ elements which leave no element in its original place. For large $p$, $!p \approx p!/e$.

[26]Similar computation is done in Appendix A.7.2 but we repeat it here to make different sections more self-contained.

[27]Unlike [30] we have $\psi_{abc}^2 = 1$.

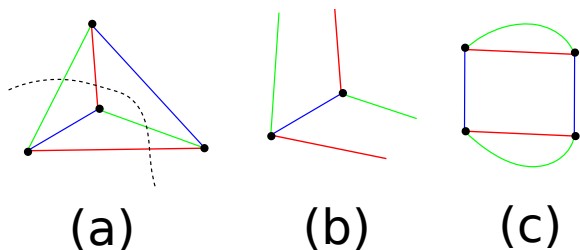

Figure 2: (a) Original CTKT hamiltonian and the cut line. (b) One $A_{bc}^{b'c'}$ part. (c) The resulting Casimir.

Diagrammatically, a singlet operator is a 3-regular graph, where vertices are fermion operators $\psi_{abc}$ and tri-colored edges are index contractions. Splitting an operator into two corresponds to a bi-partition of vertices into black and gray.

We can rewrite the Hamiltonian as:

$$H_{\text{CTKT}} = \sum_{bcb'c'} A_{bc}^{b'c'} A_{bc'}^{b'c} + \text{number of order } N^4, \tag{C.1}$$

where $A$ are (Hermitian) generators of $O(N^2)$, since we can combine a pair $(b, c)$ into a single index running from 1 to $N^2$:

$$A_{b'c'}^{bc} = \frac{i}{2} \sum_a [\psi_{abc}, \psi_{ab'c'}]. \tag{C.2}$$

Ignoring the constant in (C.1), we can rearrange the terms in $H_{CTKT}$:

$$H_{\text{CTKT}} = \frac{1}{2} \sum_{bcb'c'} \left( \left(A_{bc}^{b'c'}\right)^2 + \left(A_{bc'}^{b'c}\right)^2 - \left(A_{bc}^{b'c'} - A_{bc'}^{b'c}\right)^2 \right). \tag{C.3}$$

First two terms are Casimirs $C_2^{O(N^2)}$ of $O(N^2)$. The last term is negative so the Hamiltonian is bounded by the Casimir:

$$\langle s|H_{\text{CTKT}}|s\rangle \leq C_2^{O(N^2)}. \tag{C.4}$$

One elementary algebraic fact is that the Casimir of[28] $O(N_1)$ and the Casimir of $O(N_2 N_3)$ are related:

$$C_2^{O(N_1)} + C_2^{O(N_2 N_3)} = N_1 N_2 N_3 (N_1 + N_2 N_3 - 2). \tag{C.5}$$

Therefore the Casimirs and the Hamiltonian are bounded by $N^5$.

### C.2 Main argument

It is easy to generalized this bound for bigger singlet operators. We will work inductively in the number of fermions $2k$. We want to show that for each $2k$-fermion operator $\mathcal{O}_{2k}$ there is $2k-4$ fermion operator $\mathcal{O}_{2k-4}$ such that

$$\langle s|\mathcal{O}_{2k}|s\rangle \leq N^5 \langle s|\mathcal{O}_{2k-4}|s\rangle. \tag{C.6}$$

Consider an operator $\mathcal{O}_{2k}$ consisting of $2k$ fermions. It involves the sum over $k$ triples $(a, b, c)$. We split it into two operators:

$$\mathcal{O}_{2k} = \sum_M \mathcal{O}_k^M \tilde{\mathcal{O}}_k^M. \tag{C.7}$$

---

[28]Here we briefly return to $N_1 N_2 N_3$ notation to show how the groups are arranged.

Multi-index $M$ represents a collection of elementary indices. In Section C.3 we prove a lemma that up to $1/N$ corrections, $\mathcal{O}_k^M$ and $\widetilde{\mathcal{O}}_k^M$ are hermitian or (anti-hermitian) and they commute with each other. We estimate these $1/N$ corrections to show that they do not spoil the desired inequality (C.6). Intuitively, it happens because for small enough $k$, the sum over triples $(a, b, c)$ is dominated by configurations where all triples are different.

Without loss of generality we assume that they are both hermitian.[29] Therefore we can again use Cauchy–Schwartz inequality:[30]

$$\mathcal{O}_{2k} \leq \frac{1}{2} \sum_M \left( (\mathcal{O}_k^M)^2 + (\widetilde{\mathcal{O}}_k^M)^2 \right). \tag{C.8}$$

So in order to bound matrix elements we can cut an operator in two pieces and glue them to their respective copies.

Imagine we managed to cut in such a way that both $\mathcal{O}_k^M, \widetilde{\mathcal{O}}_k^M$ contain a $\psi$-operator with two dangling lines - Figure 3 (a). Then $(\mathcal{O}_k^M)^2$ and $\left(\widetilde{\mathcal{O}}_k^M\right)^2$ contains a bubble (C.15) which

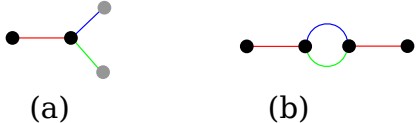

$$\text{(a)} \qquad\qquad \text{(b)}$$

Figure 3: Black and gray dots correspond to the partition into $\mathcal{O}_k^M$ and $\widetilde{\mathcal{O}}_k^M$. (a) Part of the original operator $\mathcal{O}_{2k}$. (b) What happens when we build $\left(\mathcal{O}_k^M\right)^2$.

reduces the number of $\psi$-operators to $2k-2$ with an extra $N^2$ in front. Naively, we got a even a better bound compared to (C.6):

$$\langle s | \mathcal{O}_{2k} | s \rangle \leq N^2 \langle s | \mathcal{O}_{2k-2} | s \rangle. \tag{C.9}$$

Unfortunately, unlike the original $\mathcal{O}_{2k}$, new operator $\mathcal{O}_{2k-2}$ can contain[31] $\sum_{abc} \psi_{abc} \psi_{abc} = N^3$ taking us back to the proposed bound (C.6). Indeed, it is allowed to have a $\psi_{abc} \psi_{abc}$ *after* we eliminate the new bubble. In fact, this is how a bound on Casimir operators can be obtained. So we must allow this situation. However, we must cut carefully and do not introduce an extra "stray" $\psi$ with all lines dangling, as its square will result in a extra $N^3$ from diagram on Figure 6.

For example, consider 8-fermion operator on Figure 4. After we cut as shown on the figure

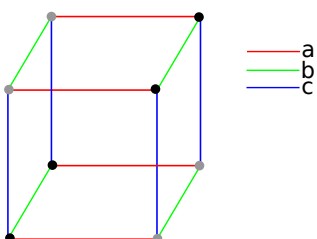

Figure 4: An example of $\mathcal{O}_8$. Black and gray dots show partition into $\mathcal{O}_4, \widetilde{\mathcal{O}}_4$

we end up with $\left( C_2^{O(N^2)} \right)^2$:

$$\mathcal{O}_8^{\text{cube}} \leq \left( C_2^{O(N^2)} \right)^2 \leq N^{10}. \tag{C.10}$$

---

[29]For anti-Hermitian operators one can multiply both of them by $i$, making them hermitian and use $-2\mathcal{O}\widetilde{\mathcal{O}} = \mathcal{O}^2 + \widetilde{\mathcal{O}}^2 - (\mathcal{O} + \widetilde{\mathcal{O}})^2$.

[30]For brevity we omit dressing the expression with $\langle \xi | \cdot | \xi \rangle$.

[31]This is discussed in detail in Section C.3.

Do we always have a proper cut? By proper we mean that both $\mathcal{O}_k^M, \widetilde{\mathcal{O}}_k^M$ have a $\psi$ with two dangling lines, and, moreover, there are no disconnected $\psi_{abc}$. We have proven by hand that such cut is possible for all connected singlet operators containing 4,8,10 fermions. We omitted 6-fermion operators, because there are no "true" six-fermion operators: all of them reduce to 4-fermion operators.

It would be convenient to contract all $a$ indices right away and so $\mathcal{O}_{2k}$ can be thought of as build from $A_{b_2 c_2}^{b_1 c_1}$:

$$A_{b_2 c_2}^{b_1 c_1} = i \sum_a \psi_{ab_1 c_1} \psi_{ab_2 c_2}, \quad (b_1 \neq b_2 \text{ or } c_1 \neq c_2). \tag{C.11}$$

One way to make sure there are no hanging $\psi_{abc}$ is to cut using $A$ only. This is possible if $k$ is even. If it is odd, we will need to cut one $A$ in half.

Let us start from proving that one can always make a proper cut if $2k \geq 12$. The proof is illustrated by Figure 5.

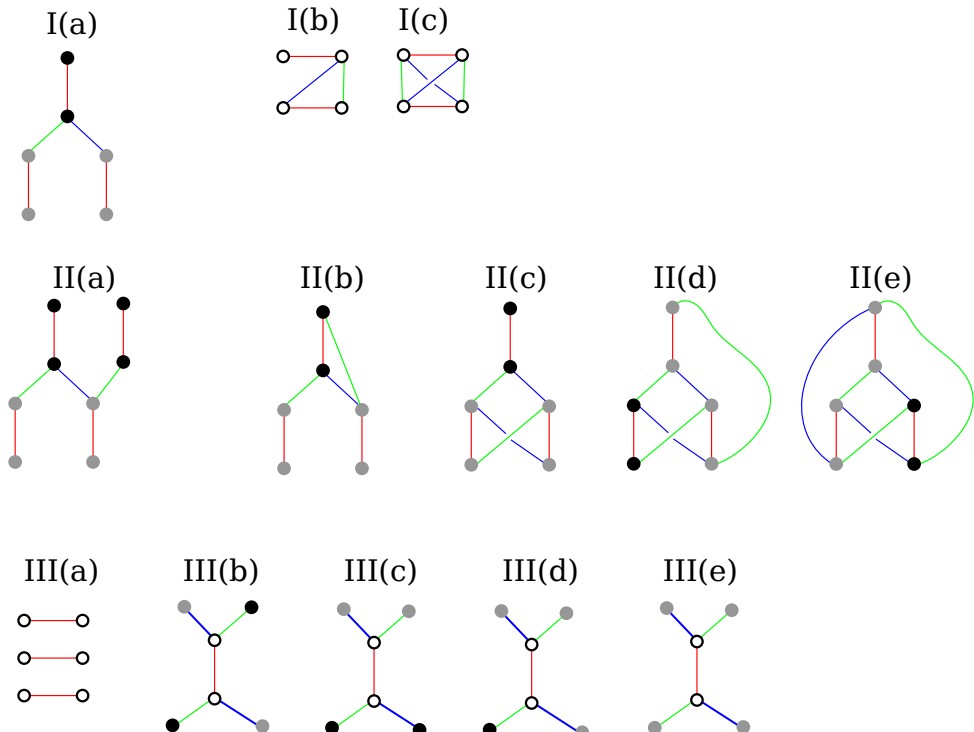

Figure 5: Three stages in constructing the proper cut.

Stage I: we pick up a $A$ combination(two $\psi$ connected by a "red" $a$ index) and declare it black. We need to consider two cases.

- I(a): Suppose one of black $\psi$ is connected to two different $A$, which we declare gray.

- I(b) Alternatively, assume that it is impossible. Then for each $A$, both of its $\psi$ are connected to the same $A$. Repeating the same argument for lower $A$ in I(b) we arrive at the Hamiltonian operator in I(c).

Stage II: consider one of the gray $\psi$ in I(a) which is connected to the initial black $A$. There are three possibilities:

- II(a): Its other leg is connected to a new $A$. We declare new $A$ to be black.

- II(b): Its other leg is connected to the initial black *A*.

- II(c): Neither of the above is true. Then the two gray *A* are connected to themselves. To be more precise, gray $\psi$, connected to the initial black *A*, have to connect to the other gray *A*. Imagine this situation happens to all triples of *A* and cases (a) and (b) never realize. Then we can re-paint the *A* and repeat the above conclusion that gray *A* are connected to themselves. This way we first arrive at II(d) and then to II(e), which is an isolated 6-fermion operator.

Cases II(a), II(b) are good, for each black and gray parts, we have a $\psi$ with two dangling indices. If *k* is even we can paint other *A* as we want. Obviously, this will not introduce "stray" $\psi$. However, if *k* is odd, we need to make sure we can cut one of the *A* in half without introducing strays.

Stage III: suppose *k* is odd. Since we are working with $2k \geq 12$, we have at least three extra *A* - III(a). We call them new *A* as in contrast to old *A* shown on II(a) and II(b). We paint all new *A* except one or two into black and gray. We do this arbitrarily, but maintaining the balance, such that in the end we need to split one remaining blank *A* into a gray or black $\psi$.

- III(b), III(c), III(d) or symmetric situations: obviously we can properly split the remaining *A*(white circles) into black and gray.

- III(e): we can not do that, since it will lead to a stray black $\psi$(there is also a symmetric situation with black and gray exchanged). To order to recover good situations like III(b,c,d) we need to change the color of at most one fermion. We can assume that all gray $\psi$ belong to the old *A* depicted on II(a) or II(b). The reason for this is that we have at least one new black *A* and one new gray *A* which we can always repaint and exchange their colors.

  Formal proof of this assumption is the following. Assume that the blank *A* is connected to one new gray *A*. This connection involves 1,2 or 3 out of 4 color lines emanating from the blank *A*. In principle it is possible that some of the gray $\psi$ in Stage III are the same. Notice that the blank *A* can not be completely connected to another *A*, as it leads to charge bubbles. We exchange this gray *A* with another black one, leading to III(b,c,d).

  The remaining step is when the blank is completely connected to the old gray *A*. In this case this choice of the blank *A* is not good and we need to pick up another one. We have at least three *A* to choose from and they can not be all connected to old gray *A*, since there are not enough color lines there.

As we briefly mentioned above, we have checked by hand that the proper cut exists for $2k = 4, 8, 10$. The above proof covers $2k \geq 12$. The proper cut always exists, so we have proven inequality (C.6). Also, as we have said before, all six-fermion operators reduce to the four-fermion operators up-to an extra $N^2$ in front. Therefore

$$\langle s|\mathcal{O}_6|s\rangle \leq N^2 \langle s|H_{\text{CTKT}}|s\rangle \leq N^7. \tag{C.12}$$

This is the origin of $1/\sqrt{N}$ for odd *k* in the bound (5.9).

## C.3   Resolving anti-commutators

Consider a singlet $\mathcal{O}_{2k}$ build out of $2k$ $\psi$-operators. We have a sum over *k* triples $I = (a, b, c)$. In this Section we argue that commuting fermions past each other will yield extra anti-commutator terms which are suppressed in *N*. This result implies right away that $\mathcal{O}_{2k}$ and its sub-parts are (anti-)Hermitian and so we can use inequality (C.8).

Let us consider case by case when indices of fermions may collide, i.e. fermion anticommutator is not trivial. Luckily, since we have Majorana fermions, collision of indices will produce operators with less fermion operators. We should keep in mind that fermionic operator with $2k$ fermions can have $k(2k-1)$ anti-commutator terms.

- Collision of $(a_1, b_1, c_1)$ and $(a_2, b_2, c_2)$.

  Suppose that in $\mathcal{O}_{2k}$ two $\psi$-operators with no common indices, e.g. $\psi_{a_1 b_1 c_1}, \psi_{a_2 b_2 c_2}$ have, in fact, the same indices. This question arises, for example, if we want to commute $\psi_{a_1 b_1 c_1}$ past $\psi_{a_2 b_2 c_2}$. It means that we substitute the product of these operators by a delta function:

  $$\{\psi_{a_1 b_1 c_1}, \psi_{a_2 b_2 c_2}\} = 2\delta_{a_1 a_2}\delta_{b_1 b_2}\delta_{c_1 c_2}. \tag{C.13}$$

  This implies that instead of $\mathcal{O}_{2k}$ we have $k(2k-1)\mathcal{O}_{2k-2}$, which is consistent with inequality (C.6) if $k \lesssim N^{5/4}$.

- Collision of $(a, b_1, c_1)$ and $(a, b_2, c_2)$.

  Now suppose that $\psi$ with one common index collide. In this case we have

  $$\sum_a \{\psi_{a b_1 c_1}, \psi_{a b_2 c_2}\} = 2N\delta_{b_1 b_2}\delta_{c_1 c_2}. \tag{C.14}$$

  So now we extra factor of in front: $N2k(k-1)\mathcal{O}_{2k-2}$. This is still consistent with (C.6) if $k \lesssim N^{3/4}$.

- Collision of $(a_1, b, c)$ and $(a_2, b, c)$.

  Since we are interested in matrix elements in singlet states, in the very beginning we can exclude the charges (5.3):

  $$\sum_{bc} \psi_{a_1 bc}\psi_{a_2 bc} = -iQ^1_{a_1 a_2} + N^2\delta_{a_1 a_2}. \tag{C.15}$$

  Since we are interested in the singlet states, the charge operator can be commuted to bra or ket, producing a number of the order of the number of fermions and reducing the number of operators from $2k$ to $2k-2$. So both terms effectively reduce the number of fermions to $2k-2$. We will assume that we have an operator build from no more than $N^2$ fermions. This way the second part dominates. It means that instead of bounding $\mathcal{O}_{2k}$ we have to bound

  $$kN^2\mathcal{O}_{2k-2}. \tag{C.16}$$

  This is a factor of $k$ because there are at most $k$ such pairs with two indices contracted. This is consistent with eq. (C.6) if $k \le N^{1/2}$.

- Collision of $(a, b, c)$ and $(a, b, c)$ This is a drastic situation when $\mathcal{O}_{2k}$ contains a disconnected piece - Figure 6

  $$\sum_{abc} \psi_{abc}\psi_{abc} = N^3. \tag{C.17}$$

  In fact, this is not possible. In the original problem of decomposing a product of $\psi$

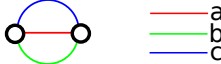

Figure 6: Diagrammatic representation for $\sum_{abc} \psi_{abc}\psi_{abc}$.

into singlets such combinations are absent. Indeed, if there is two identical $\psi_{abc}$, their

product can be substituted by 1 and we need to decompose a product of $2k-2$ fermions into singlets. In our manipulations with the operators in the main part of the proof, we made sure not to introduce such disconnected factor. We might worry that while computing anti-commutators, when we substitute a product $\psi\psi$ by a delta function, we might accidentally create such disconnected pieces. But this is actually not possible because $\psi$ share at most one common index as we have explained above.

# D An argument for 't Hooft scaling

Let us argue why 't Hooft scaling holds in the low energy sector of BFSS.

Field content of BFSS/BMN has 16 (spinor of $SO(9)$) real Majorana fermions $\psi^a$, $a = 1, \ldots, 16$, and 9 (vector of $SO(9)$) real scalars $X^\mu$ both in the adjoint representation of $SU(N)$. BFSS model is essentially a dimensional reduction of $\mathcal{N} = 4$ $SU(N)$ super Yang-Mills theory from $3+1$ dimensions to one dimension. BMN is a special massive deformation. Yang-Mills coupling constant $g_{YM}^2$ in 1 dimension has dimension of mass$^3$. Therefore the system is strongly coupled in the IR. In this regime, when the 't Hooft constant $\lambda = g_{YM}^2 N$ is large BFSS is conjectured to be dual to Einstein–dilaton gravity in 10 dimensions [58]. However, since this theory does not have conformal symmetry, the geometry is not $AdS_2 \times S^8$: the $S^8$ radius is not constant and there is a large curvature region corresponding to the UV regime where 't Hooft coupling is small.

The Lagrangian of BFSS is the dimension reduction of four-dimensional $\mathcal{N} = 4$ $SU(N)$ super Yang–Mills to one dimension:

$$\mathcal{L} = \frac{1}{g_{YM}^2} \text{Tr}\left( \frac{1}{2} \sum_{\nu=1}^{9} (D_t X^\nu)^2 + \frac{1}{2} \chi D_t \chi + \frac{1}{4} [X^\nu, X^\eta]^2 + i \frac{1}{2} \chi \gamma^\nu [\chi, X^\nu] \right), \tag{D.1}$$

where $\gamma^\nu$ are 10-dimensional Gamma-matrices $D_t = \partial_t + i[A_t, \cdot]$ is the covariant derivative in the adjoint representation. There is one coupling in this model: $g_{YM}$ is the gauge coupling constant.

First of all, let us explain why the power $5/2$ in (1.7) is consistent with 't Hooft scaling. Naively, one expects a factor of $N$ for each trace. However, in our case fermion operators are normalized differently. In the 't Hooft argument one considers Lagrangian in the form

$$\mathcal{L} = \frac{N}{\lambda} \text{Tr}\left( (DX)^2 + \chi D \chi + \ldots \right), \tag{D.2}$$

where we have omitted the interaction terms, and $X$ and $\chi$ schematically refer to a collection of scalar and fermionic fields. In this normalization, inspecting Feynman diagrams one indeed concludes that single-trace expectation values of $\chi$ scale as $N$

$$\langle \text{Tr}(\chi^k) \rangle \sim N. \tag{D.3}$$

However, $\chi$ have non-canonical commutation relations, because of the factor $N/\lambda$ in front of the Lagrangian: $\chi^2 \sim \frac{\lambda}{N}$. In the present paper we work with fermions in the canonical normalization $\psi^2 \sim 1$, hence $\psi \propto \chi \sqrt{N}$. Hence for canonically normalized fermions

$$\langle \text{Tr}(\psi^k) \rangle \sim N^{1+k/2}. \tag{D.4}$$

However, in quantum mechanics we solve Schrödinger equation instead of computing Feynman diagrams. So it is not obvious that 't Hooft scaling should hold. However, in case of BFSS model it was argued that it holds at least for simple expectation values such as $\text{Tr}(X^1)^2$.

Using supersymmetry and virial theorem, one can rigorously show [59] that the "typical size" $R$ of the vacuum state is

$$R^2 = \langle 0| \operatorname{Tr}\left[(X_1)^2\right]|0\rangle \gtrsim N\lambda^{2/3}. \tag{D.5}$$

It is believed that this inequality is saturated. For example, one can study non-singlet excitations in this model [28]. In can be shown that the energy gap between the vacuum and the lightest $SU(N)$-charged(adjoint) state is

$$E_{\mathrm{adj}} \propto \frac{N\lambda}{R^2}. \tag{D.6}$$

't Hooft scaling (D.5) implies that non-singlets are separated by a finite gap from the ground state. Monte–Carlo simulations support this conjecture [29].

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
