# Peer review of "Quantum error correction and large $N$"

_SciPost Physics, doi:SciPost Phys. 11, 094 (2021)_

## Round 2 · Referee Report · Anonymous (Referee 1) · 2021-8-12

Report
I think this is a terrific and important paper. The work is extremely well-motivated: despite all the talk about error correcting codes and holography, before this work it was not at all clear how to see a priori that a large-$N$ gauge theory has something to do with error correction. Indeed one might have thought that the intricate and horrible details of the particular hamiltonians admitting simple gravity duals (e.g. a sparse spectrum) might have been required. But this paper makes it very clear that none of that is necessary, and that the requisite error correction property follows purely from kinematics of large $N$ singlets.
Furthermore, the paper gives a quantitative analysis of how the error correction property breaks down at finite $N$.
Since this property seems to be a deep fact about gravity, I believe this result will be quite important.
Indeed the main result of the paper implies that the error correction property transcends semiclassical gravity and applies to any semiclassical string theory.
The paper is also well written, with a very readable and compelling introduction.
One vague complaint, which actually can be blamed on ref [27]: there is essentially no such thing as "gauge symmetry", only gauge redundancy, and the presence of such a redundancy is not a physical property (for example, different dual descriptions of the same system need not have the same gauge group). The statement that "QEC is tied to the presence of gauge symmetries" is therefore a problematic one. It would be nice to know what exactly is its invariant meaning.
Having said all these positive things I must admit that I have a confusion about the main premise.
The basic effect demonstrated here is that non-gauge invariant states (at least in these models with adjoint fermions) are approximately orthogonal at large $N$.
If, as in the toric code, we regard such "charged" (or "colored") excitations as errors, the fact that they are orthogonal means that different errors can be distinguished and therefore corrected.
The toric code is a UV completion of a (discrete) gauge theory, where such failures of gauge invariance occur only at high energy.
From this point of view of emergent gauge theory, the notion of the singlet sector as an error-correcting code makes perfect sense.
In the context of gauge theories with gravity duals, however, we usually regard the gauge invariance as an exact redundancy, so that the physical hilbert space contains only gauge invariant states. In that context, I do not understand a physical role for these states with errors.
-- "spacial" should be "spatial"
-- "so the are not dynamical"
should be "so they are not dynamical"
-- "can be draw from one fixed set of indices"
should be "can be drawn from one fixed set of indices"
Furthermore, the paper gives a quantitative analysis of how the error correction property breaks down at finite $N$.
Since this property seems to be a deep fact about gravity, I believe this result will be quite important.
Indeed the main result of the paper implies that the error correction property transcends semiclassical gravity and applies to any semiclassical string theory.
The paper is also well written, with a very readable and compelling introduction.
One vague complaint, which actually can be blamed on ref [27]: there is essentially no such thing as "gauge symmetry", only gauge redundancy, and the presence of such a redundancy is not a physical property (for example, different dual descriptions of the same system need not have the same gauge group). The statement that "QEC is tied to the presence of gauge symmetries" is therefore a problematic one. It would be nice to know what exactly is its invariant meaning.
Having said all these positive things I must admit that I have a confusion about the main premise.
The basic effect demonstrated here is that non-gauge invariant states (at least in these models with adjoint fermions) are approximately orthogonal at large $N$.
If, as in the toric code, we regard such "charged" (or "colored") excitations as errors, the fact that they are orthogonal means that different errors can be distinguished and therefore corrected.
The toric code is a UV completion of a (discrete) gauge theory, where such failures of gauge invariance occur only at high energy.
From this point of view of emergent gauge theory, the notion of the singlet sector as an error-correcting code makes perfect sense.
In the context of gauge theories with gravity duals, however, we usually regard the gauge invariance as an exact redundancy, so that the physical hilbert space contains only gauge invariant states. In that context, I do not understand a physical role for these states with errors.
-- "spacial" should be "spatial"
-- "so the are not dynamical"
should be "so they are not dynamical"
-- "can be draw from one fixed set of indices"
should be "can be drawn from one fixed set of indices"

---

## Round 3 · Author Response

Dear Referee,
Thank you very much for your constructive comments.

>The statement that "QEC is tied to the presence of gauge symmetries" is therefore a problematic one. It would be >nice to know what exactly is its invariant meaning.
As far as I understand the results of ref. [27], it was the presence of gauge redundancy that allowed them to build precursor operators. In other, more invariant words, they linked the redundancy of quantum error correction to the gauge redundancy. I changed the corresponding statement in the Introduction as "It was suggested earlier
[27] that the redundancy in holographic QEC maybe tied the gauge redundancy."

>In the context of gauge theories with gravity duals, however, we usually regard the gauge invariance as an exact >redundancy, so that the physical hilbert space contains only gauge invariant states. In that context, I do not >understand a physical role for these states with errors.
This is a very good comment and I completely agree with it. From bigger perspective I see the current results just as a first step in understanding more general quantum error correcting properties. I have been thinking about more general setup, where errors are allowed to be singlets, but I have not made much progress.
As a "consolation prize", the results in the paper can be helpful for simple matrix models such as BFSS. Ref. [28] argued that ungauging SU(N) in BFSS preserves the bulk dual. Non-singlet states correspond to folded strings in the bulk. Also ref. [28] argued that such non-singlet excitations always have a large energy and so they are localized near the boundary. The argumentation was based on bulk locality and is not directly applicable to other matrix models without smooth bulk dual. However, ref. [29] observed numerically that in other "ungauged" matrix models non-singlets have high energy too. Results in current paper might suggest that this is a general phenomena at large N. However, I have not managed to formulate this more rigorously, as properties of near-orthogonality and being highly energetic are not necessarily related.

Best regards,
Alexey

---

## Editorial Decision

published